# The Polynomial Stein Discrepancy for Assessing Moment Convergence

**Narayan Srinivasan** [1,2]  **Matthew Sutton** [3]  **Christopher Drovandi** [1,2]  **Leah South** [1,2]

## Abstract

We propose a novel method for measuring the discrepancy between a set of samples and a desired posterior distribution for Bayesian inference. Classical methods for assessing sample quality like the effective sample size are not appropriate for scalable Bayesian sampling algorithms, such as stochastic gradient Langevin dynamics, that are asymptotically biased. Instead, the gold standard is to use the kernel Stein Discrepancy (KSD), which is itself not scalable given its quadratic cost in the number of samples. The KSD and its faster extensions also typically suffer from the curse-of-dimensionality and can require extensive tuning. To address these limitations, we develop the polynomial Stein discrepancy (PSD) and an associated goodness-of-fit test. While the new test is not fully convergence-determining, we prove that it detects differences in the first $r$ moments for Gaussian targets. We empirically show that the test has higher power than its competitors in several examples, and at a lower computational cost. Finally, we demonstrate that the PSD can assist practitioners to select hyper-parameters of Bayesian sampling algorithms more efficiently than competitors.

## 1. Introduction

Reliable assessment of posterior approximations was recently named one of three "grand challenges in Bayesian computation" (Bhattacharya et al., 2024). Stein discrepancies (Gorham & Mackey, 2015) can assist with this by providing ways to assess whether a distribution $Q$, known via samples $\{x^{(i)}\}_{i=1}^{n} \sim Q$, is a good approximation to a target distribution $P$. These discrepancies can be applied when $P$ is only known up to a normalising constant, making them particularly appealing in the context of Bayesian inference. Unlike classical tools for assessing sample quality in Bayesian inference, Stein discrepancies are applicable to hyperparameter tuning (Gorham & Mackey, 2017) and goodness-of-fit testing (Chwialkowski et al., 2016; Liu et al., 2016) for biased Monte Carlo algorithms like stochastic gradient Markov chain Monte Carlo (SG-MCMC, Welling & Teh, 2011; Nemeth & Fearnhead, 2021).

Currently, the most widely adopted Stein discrepancy is the kernel Stein discrepancy (KSD, Gorham & Mackey, 2017; Chwialkowski et al., 2016; Liu et al., 2016), which applies a so-called Stein operator (Stein, 1972) to a reproducing kernel Hilbert space. The KSD has an analytically tractable form and can perform well in tuning and goodness-of-fit tasks using the recommended inverse multiquadric (IMQ) base kernel. However, the computational cost of KSD is quadratic in the number of samples, $n$, from the distribution $Q$, so it is infeasible to directly apply KSD in applications where large numbers of MCMC iterations have been used. To address this bottleneck, Liu et al. (2016) introduced an early linear-time alternative. However, the statistic suffers from poor statistical power compared to its quadratic-time counterpart, meaning that the probability of rejecting the null hypothesis when it is false is low. Several approaches have been disseminated in the literature to tackle these problems and reduce the cost of using Stein discrepancies. Notably, the finite set Stein discrepancy (FSSD, Jitkrittum et al., 2017) and the random feature Stein discrepancy (RFSD, Huggins & Mackey, 2018) have been proposed as linear-time alternatives to KSD.

The main idea behind FSSD is to use a finite set of test locations to evaluate the Stein witness function, which can be computed in linear-time. The FSSD test can effectively capture the differences between $P$ and $Q$, by optimizing the test locations and kernel bandwidth. However, the test is sensitive to the test locations and other tuning parameters, all of which need to be optimized (Jitkrittum et al., 2017). The test requires that the samples be split for this optimisation and for evaluation. Additionally, the FSSD experiences a degradation of power relative to the standard KSD in high dimensions (Huggins & Mackey, 2018).

[1]School of Mathematical Sciences, Queensland University of Technology (QUT), Brisbane, Australia [2]QUT Centre for Data Science, Brisbane, Australia [3]School of Mathematics and Physics, University of Queensland, Brisbane, Australia. Correspondence to: Narayan Srinivasan <narayanseshadri.srinivasan@hdr.qut.edu.au>, Leah South <l1.south@qut.edu.au>.

*Proceedings of the 42$^{nd}$ International Conference on Machine Learning*, Vancouver, Canada. PMLR 267, 2025. Copyright 2025 by the author(s).

Random Fourier features (RFF) (Rahimi & Recht, 2007) are a well-established technique to accelerate kernel-based methods. However, it is known (Chwialkowski et al., 2015, Proposition 1) that the resulting statistic fails to distinguish a large class of probability measures. Huggins & Mackey (2018) alleviate this by generalising the RFF approach with their RFSD method, which is near-linear-time. The authors identify a family of convergence-determining discrepancy measures that can be accurately approximated with importance sampling. Like FSSD, tuning can be challenging for RFSD because of the large number of example-specific choices, including the optimal feature map, the importance sampling distribution, and the number of importance samples. We have also found empirically that the power of the goodness-of-fit test based on RFSD is reduced when direct sampling from $P$ is infeasible.

In addition to these limitations, KSD and current linear-time approximations can fail to detect moment convergence (Kanagawa et al., 2022). The standard Langevin KSD using the IMQ kernel controls the bounded-Lipschitz metric, which determines weak convergence, but fails to control the convergence in moments. This is a significant drawback because moments are often the main expectations of interest and, for many biased MCMC algorithms, they are where bias is likely to appear, as explained in Section 3.3. Kanagawa et al. (2022) propose an extended, quadratic-time KSD that is able to control convergence in moments but this method is slower than KSD with an IMQ kernel.

Motivated by the shortcomings of KSD and linear-time KSD methods, we propose a linear-time variant of the KSD named the polynomial Stein Discrepancy (PSD). The method we propose in this article detects discrepancies in moments while still being computable in linear time. This approach, based on the use of $r$th order polynomials, is motivated by the zero variance control variates (Assaraf & Caffarel, 1999; Mira et al., 2013) used in variance reduction of Monte Carlo estimates. While PSD is not fully convergence-determining, we show that when $P$ is Gaussian, which is a reasonable approximation for big data applications (Bardenet et al., 2017), the discrepancy is zero if and only if the first $r$ moments of $P$ and $Q$ match. We empirically show that PSD has good statistical power for detecting discrepancies in moments, including in applications with non-Gaussian $P$, and we also demonstrate its effectiveness for tuning SG-MCMC algorithms. Importantly, the decrease in power with increasing dimension is considerably less than competitors, and the method requires no calibration beyond the choice of the polynomial order, which has a clear interpretation. Liu & Wang (2018) demonstrate the theoretical advantages of using similar kernels for moment matching in Stein variational gradient descent.

The paper is organized as follows. Section 2 sets notation and provides background on Stein discrepancies, KSD and goodness-of-fit tests. The PSD and goodness-of-fit tests based on the same are presented in Section 3, along with theoretical results about detecting moment discrepancies and the asymptotic power of the test. Section 4 contains simulation studies performed on benchmark examples. The paper is concluded in Section 5.

## 2. Background

Let the probability measure $Q$, known through the $n$ samples, $\{x^{(i)}\}_{i=1}^n$, be supported in $\mathcal{X}$. Suppose the target distribution $P$ has a corresponding probability density function $p$.

Utilising the notion of integral probability metrics (IPMs, Müller, 1997), a discrepancy between $P$ and $Q$ can be defined as

$$d_{\mathcal{H}}(P,Q) := \sup_{h \in \mathcal{H}} |\mathbb{E}_{X \sim P}[h(X)] - \mathbb{E}_{X \sim Q}[h(X)]|. \quad (1)$$

Various IPMs correspond to different choices of function class $\mathcal{H}$ (Anastasiou et al., 2023). However, since we only have access to the unnormalized density $P$, the first expectation in (1) is typically intractable.

### 2.1. Stein Discrepancies

Stein discrepancies make use of a so-called Stein operator ($\mathcal{A}$) and an associated class of functions, $\mathcal{G}(\mathcal{A})$, such that

$$\mathbb{E}_{X \sim P}[(\mathcal{A}g)(X)] = 0 \text{ for all } g \in \mathcal{G}(\mathcal{A}). \quad (2)$$

This allows us to define the so-called Stein discrepancy

$$\mathcal{S}(Q, \mathcal{A}, \mathcal{G}) = \sup_{g \in \mathcal{G}(\mathcal{A})} \| \mathbb{E}_{X \sim Q}[(\mathcal{A}g)(X)] \|. \quad (3)$$

There is flexibility in the choice of Stein operator and we focus on operators that are appropriate when $x \in \mathcal{X} \subseteq \mathbb{R}^d$. By considering the generator method of Barbour (1990) applied to overdamped Langevin diffusion (Roberts & Tweedie, 1996), one arrives (Gorham & Mackey, 2015) at the second order Langevin-Stein operator defined for real-valued $g$ as

$$\mathcal{A}_x^{(2)}g = \triangle_x g(x) + \nabla_x g(x) \cdot \nabla_x \log p(x). \quad (4)$$

Under mild conditions on $p$ and $g$, $\mathbb{E}_{X \sim P}[(\mathcal{A}_x^{(2)}g)(X)] = 0$ as required. For example for unconstrained spaces where $\mathcal{X} = \mathbb{R}^d$, if $g$ is twice continuously differentiable, $\log p$ is continuously differentiable and $\|\nabla g(x)\| \leq C\|x\|^{-\delta}p(x)^{-1}$, for some constant $C > 0$ and $\delta > 0$, then $\mathbb{E}_{X \sim P}[(\mathcal{A}_x^{(2)}g)(x)] = 0$ as required (South et al., 2022).

Alternatively, one can consider a first order Langevin-Stein operator defined for vector-valued $g$ (Gorham & Mackey,

2015)

$$\mathcal{A}_x^{(1)} g = \nabla_x \cdot g(x) + g(x) \cdot \nabla_x \log p(x). \quad (5)$$

Under regularity conditions on $g$ and $p$ similar to those for (4) we get that, $\mathbb{E}_P[(\mathcal{A}_x^{(1)} g)(X)] = 0$.

## 2.2. Kernel Stein Discrepancy

The key idea behind KSD is to write the maximum discrepancy between the target distribution and the observed sample distribution by considering $\mathcal{G}$ in (3) to be functions in a reproducing kernel Hilbert space (RKHS, Berlinet & Thomas-Agnan, 2004) corresponding to an appropriately chosen kernel.

A symmetric, positive-definite kernel $k : \mathbb{R}^d \times \mathbb{R}^d \to \mathbb{R}$ induces an RKHS $\mathcal{K}_k$ of functions from $\mathbb{R}^d \to \mathbb{R}$. For any $x \in \mathbb{R}^d$, $k(x, \cdot) \in \mathcal{K}_k$. From the reproducing property of the RKHS, if $f \in \mathcal{K}_k$ then $f(x) = \langle f, k(x, \cdot) \rangle$.

The Stein set of functions $\mathcal{G}$ with the associated kernel is taken to be the set of vector-valued functions $g$, such that each component $g_j$ belongs to $\mathcal{K}_k$ and the vector of their norms $\|g_j\|_{\mathcal{K}_k}$ belongs to the unit ball, i.e. $\mathcal{G} := \{g = (g_1, \ldots, g_d) : \|v\| \leq 1 \text{ for } v_j = \|g_j\|_{\mathcal{K}_k}\}$.

Under certain regularity conditions enforced on the choice of the kernel $k$, Gorham & Mackey (Proposition 2, 2017) and also Liu et al. (2016); Chwialkowski et al. (2016) arrive at the closed form representation for the Stein discrepancy given in (3) for this particular choice of Stein set and label it the KSD

$$\text{KSD} := \mathcal{S}(Q, \mathcal{A}, \mathcal{G}) = \sqrt{\mathbb{E}_{x, x' \sim Q}[k_0(x, x')]}, \quad (6)$$

where in the particular case of the first order operator (5),

$$k_0(x, x') = \nabla_x \cdot \nabla_x' k(x, x') + \nabla_x k(x, x') \cdot u(x') \\ + \nabla_{x'} k(x, x') \cdot u(x) + k(x, x') u(x) \cdot u(x').$$

Here, $u(x) = \nabla_x \log p(x)$ and $k(x, x')$ is the chosen kernel.

Gorham & Mackey (Theorem 6, 2017) show that KSDs based on common kernel choices like the Gaussian kernel $k(x, x') = \exp(-\frac{1}{2h^2} \|x - x'\|_2^2)$ and the Matérn kernel fail to detect non-convergence for $d \geq 3$. Gaussian kernels are also known to experience rapid decay in statistical power in increasing dimensions for common errors (Gorham & Mackey, 2017). As an alternative, Gorham & Mackey (2017) recommend the IMQ kernel $k(x, x') = (c^2 + \|x - x'\|_2^2)^\beta$ with $c = 1$ and $\beta = -0.5$. They show that IMQ KSD detects convergence and non-convergence for $c > 0$ and $\beta \in (-1, 0)$ for the class of distantly dissipative $P$ with Lipschitz $\log p$ and $\mathbb{E}_{x \sim P}[\|\nabla_x \log p(x)\|_2^2] < \infty$,

and they provide a lower-bound for the KSD in terms of the bounded Lipschitz metric.

The discrepancy $\mathcal{S}(Q, \tau, \mathcal{G})$ can be estimated with its corresponding U-statistic (Serfling, 2009) as follows

$$\widehat{\text{KSD}}^2 = \frac{1}{n(n-1)} \sum_{i=1}^n \sum_{\substack{j=1 \\ j \neq i}}^n k_0(x^{(i)}, x^{(j)}). \quad (7)$$

This is the minimum variance unbiased estimator of KSD (Liu et al., 2016). Alternatively, one could consider the V-Statistic proposed in Gorham & Mackey (2017) and Chwialkowski et al. (2016), given by

$$\widehat{\text{KSD}}^2 = \frac{1}{n^2} \sum_{i=1}^n \sum_{j=1}^n k_0(x^{(i)}, x^{(j)}). \quad (8)$$

The V-statistic, while no longer unbiased, is strictly non-negative and hence can be better suited as a discrepancy metric when compared to the U-statistic.

## 2.3. Goodness-of-Fit Testing

In goodness-of-fit testing, the objective is to test the null hypothesis $H_0$: $Q = P$ against an alternative hypothesis, typically that $H_1$: $Q \neq P$. Existing goodness-of-fit tests are based on either asymptotic distributions of U-statistics (Liu et al., 2016; Jitkrittum et al., 2017; Huggins & Mackey, 2018) or bootstrapping (Liu et al., 2016; Chwialkowski et al., 2016).

Asymptotic goodness-of-fit tests cannot be implemented directly for KSD or its early linear-time alternatives (Liu et al., 2016; Chwialkowski et al., 2016). Instead, bootstrapping is used to estimate the distribution of the test statistic under $H_0$. Chwialkowski et al. (2016) develop a test for potentially correlated samples $\{x^{(i)}\}_{i=1}^n$ using the wild bootstrap procedure (Leucht & Neumann, 2013). Liu et al. (2016) use the bootstrap procedure of Hušková & Janssen (1993); Arcones & Gine (1992) for degenerate U-statistics, in the setting where the samples are uncorrelated. As the number of samples $n$ and the number $m$ of bootstrap samples go to infinity, Liu et al. (2016) and Chwialkowski et al. (2016) show[1] that their bootstrap methods have correct type I error rate (i.e. correct probability of rejecting the null hypothesis when it is true) and power one.

More recent linear-time alternatives, specifically the FSSD and RFSD, employ tests based on the asymptotic distribution of U-statistics. These methods have similar theoretical guarantees as $n \to \infty$ and the number $m$ of simulations from the tractable null distribution go to infinity but they

---

[1]Liu et al. (2016) do not appear to directly state the requirement that $n \to \infty$ in their theorem, but this follows from the proof on which it is based (Hušková & Janssen, 1993).

can suffer from poor performance in practice. These tests require an estimate for a covariance under $P$. When direct sampling from $P$ is infeasible, which is typically the case when measuring sample quality for Bayesian inference, these methods estimate this covariance using samples from $Q$. While this is asymptotically correct as $n \to \infty$ (Jitkrittum et al., 2017; Huggins & Mackey, 2018), we have found empirically that the use of samples from $Q$ can substantially reduce the statistical power of the tests, particularly for RFSD.

## 3. Polynomial Stein Discrepancy

Motivated by the practical effectiveness of polynomial functions in MCMC variance reduction and post-processing (Mira et al., 2013; Assaraf & Caffarel, 1999), we develop PSD as a linear-time alternative to KSD.

### 3.1. Formulation

This section presents an intuitive and straightforward approach to derive the PSD. The corresponding goodness-of-fit tests are presented in Section 3.2.

Consider the class of $r$th order polynomials. That is, let $\mathcal{G} = \text{span}\{\prod_{i=1}^{d} x[i]^{\alpha_i} : \alpha \in \mathbb{N}_0^d, \sum_{i=1}^{d} \alpha_i \leq r\}$, where $x[i]$ denotes the $i$th dimension of $x$. Intuitively, $\mathcal{G}$ is the span of $J = \binom{d+r}{d} - 1$ monomial terms that we will simply denote by $P_i(x)$ for $i = 1, \ldots, J$. This is a valid Stein set in that $\mathbb{E}_P[\mathcal{A}_x^{(2)} g(x)] = 0$ for all $g \in \mathcal{G}$ under mild conditions depending on the distribution $P$. For example, when the density of $P$, $p(x)$, is supported on an unbounded set then a sufficient condition is that the tails of $P$ decay faster than an $r$th order polynomial. One can also consider boundary conditions for bounded spaces, as per Proposition 2 of Mira et al. (2013).

Henceforth we denote the second order operator $\mathcal{A}_x^{(2)}$ as simply $\mathcal{A}$. Using the linearity of the Stein operator (4), the aim is to optimize over different choices $\beta$ with real coefficients $\beta_k$ for $k = 1, 2, \ldots, J$ in (3). Analogous to the optimization in KSD, the optimization is constrained over the unit ball, that is $\|\beta\|_2^2 \leq 1$. The result is

$$
\begin{aligned}
\text{PSD} &= \sup_{g \in \mathcal{G}} |\mathbb{E}_Q[\mathcal{A}g(x)]| \\
&= \sup_{\beta \in \mathbb{R}^J : \|\beta\|_2 \leq 1} \left| \mathbb{E}_Q \left[ \sum_{k=1}^{J} \beta_k \mathcal{A} P_k(x) \right] \right| \\
&= \sup_{\beta \in \mathbb{R}^J : \|\beta\|_2 \leq 1} \left| \sum_{k=1}^{J} \beta_k \mathbb{E}_Q [\mathcal{A} P_k(x)] \right| \\
&= \sqrt{\sum_{k=1}^{J} \bar{z}_k^2},
\end{aligned} \tag{9}
$$

where $\bar{z}_k = \mathbb{E}_Q[\mathcal{A} P_k(X)]$. This derivation is provided in more detail in Appendix A. For the sample $\{x^{(i)}\}_{i=1}^n$ from $Q$, we have $\bar{z}_k = \frac{1}{n} \sum_{i=1}^{n} \mathcal{A} P_k(x^{(i)})$. Note that PSD implicitly depends on a polynomial order $r$ through $J$. A simple form for $\mathcal{A} P_k(x)$ is available in Appendix A of South et al. (2023).

The squared linear-time solution for PSD can also be expressed as a V-statistic. Specifically

$$
\begin{aligned}
\widehat{\text{PSD}}_v^2 &= \sum_{k=1}^{J} \bar{z}_k^2 \\
&= \frac{1}{n^2} \sum_{i=1}^{n} \sum_{j=1}^{n} \Delta(x^{(i)}, x^{(j)}),
\end{aligned}
$$

where $\Delta(x, y) = \tau(x)^\top \tau(y) = \sum_{k=1}^{J} \mathcal{A}_x P_k(x) \mathcal{A}_y P_k(y)$ and $[\tau(x)]_k = \mathcal{A} P_k(x)$.

The U-statistic version of the squared PSD, which will be helpful in goodness-of-fit testing, is

$$
\begin{aligned}
\widehat{\text{PSD}}_u^2 &= \frac{1}{n(n-1)} \sum_{i=1}^{n} \sum_{\substack{j=1 \\ j \neq i}}^{n} \Delta(x^{(i)}, x^{(j)}) \\
&= \frac{1}{n(n-1)} \left( n^2 \sum_{k=1}^{J} \bar{z}_k^2 - n \sum_{k=1}^{J} \overline{z_k^2} \right),
\end{aligned} \tag{10}
$$

where $\overline{z_k^2} = \frac{1}{n} \sum_{i=1}^{n} \left( \mathcal{A} P_k(x^{(i)}) \right)^2$.

The computational complexity of this discrepancy is $\mathcal{O}(nJ) = \mathcal{O}(n\binom{d+r}{d})$. In very high dimensions, practitioners concerned about computational cost could run an approximate version of the test for $\mathcal{O}(ndr)$ by excluding interaction terms from the polynomial, for example for $r = 2$, monomial terms $x_i x_j$ would only be included for $i = j$. In the case of a Gaussian $P$ with diagonal covariance matrix, such a discrepancy would detect differences in the marginal moments.

We note that while it would be possible to implement KSD with the conventional polynomial kernel $k(x, y) = (1 + x^\top y)^r$, such discrepancies have not yet been extensively explored in the literature. We show the promise of polynomial kernels in terms of statistical power and linear-time complexity. Our PSD also differs from what one would obtain with a conventional polynomial kernel, offering a simpler formulation that may be more effective for identifying specific moments where discrepancies occur; in the case of a Gaussian $P$ with independent components, monomial terms correspond directly to multi-index moments. We also present novel theory (Proposition 3.2) that enhances the understanding and interpretability of PSD in the Bayesian big data limit.

## 3.2. Goodness-of-fit Test

For the goodness-of-fit test based on PSD, we will be testing the null hypothesis $H_0: Q = P$ against a *composite* or *directional* alternative hypothesis. The form of the alternative hypothesis depends on $P$, but we show in Section 3.3 that for Gaussian P(a suitable assumption under the Bayesian big data/Bernstein-Von-Mises limit), $H_1$ is that the first $r$ moments of $Q$ do not match the first $r$ moments of $P$. Achieving high statistical power for these moments is important, as described in Section 3.3.

Observe that (10) is a degenerate U-statistic, so an asymptotic test can be developed using $[\tau(x)]_k = \mathcal{A}P_k(x)$.

**Corollary 3.1** (Jitkrittum et al. (2017)). *Let $Z_1, \ldots, Z_J$ be i.i.d. random variables with $Z_i \sim \mathcal{N}(0,1)$. Let $\mu := \mathbb{E}_{x \sim Q}[\tau(x)]$ and $\Sigma_r := cov_{x \sim r}[\tau(x)] \in \mathbb{R}^{J \times J}$ for $r \in \{P, Q\}$. Let $\{\omega_i\}_{i=1}^J$ be the eigenvalues of the covariance matrix $\Sigma_p = \mathbb{E}_{x \sim p}[\tau(x)\tau^\top(x)]$. Assume that $\mathbb{E}_{x \sim Q}\mathbb{E}_{y \sim Q}\Delta^2(x, y) < \infty$. Then, the following statements hold*

1. *Under $H_0: Q = P$,*

$$n\widehat{PSD}_u^2 \xrightarrow{d} \sum_{i=1}^J (Z_i^2 - 1)\omega_i.$$

2. *Under $H_1$ ($Q \neq P$ in a way specified by PSD), if $\sigma_{H1}^2 := 4\mu^\top \Sigma_q \mu > 0$, then*

$$\sqrt{n}(\widehat{PSD}_u^2 - PSD^2) \xrightarrow{d} \mathcal{N}(0, \sigma_{H1}^2).$$

*Proof.* This follows immediately from Proposition 2 of Jitkrittum et al. (2017), which itself follows from Liu et al. (Theorem 4.1, 2016) and Serfling (Chapter 5.5, 2009). □

A simple approach to testing $H_0: Q = P$ would then be to simulate from the stated distribution under the null hypothesis $m$ times and to reject $H_0$ at a significance level of $\alpha$ if the observed test statistic (10) is above the $T_\alpha = 100(1 - \alpha)$ percentile of the $m$ samples. However, simulating from this distribution can be impractical since $\Sigma_p$ requires samples from the target $P$, which is typically not feasible. Following the suggestion of Jitkrittum et al. (2017), $\Sigma_p$ can be replaced with the plug-in estimate computed from the sample, $\hat{\Sigma}_q := \frac{1}{n}\sum_{i=1}^n \tau(x_i)\tau(x_i)^\top - \left[\frac{1}{n}\sum_{i=1}^n \tau(x_i)\right]\left[\frac{1}{n}\sum_{j=1}^n \tau(x_j)\right]^\top$. Theorem 3 of Jitkrittum et al. (2017) ensures that replacing the covariance matrix with $\hat{\Sigma}_q$ still renders a consistent test.

In practice, however, there may be unnecessary degradation of power, especially in high dimensions due to the approximation of the covariance in Corollary 3.1. To circumvent this, we propose an alternative test which is to follow the

bootstrap procedure suggested by Liu et al. (2016), Arcones & Gine (1992), Hušková & Janssen (1993). This bootstrap test will itself be linear-time; such a method was not available in the linear-time KSD alternatives of Jitkrittum et al. (2017) and Huggins & Mackey (2018).

In each bootstrap replicate of this test, one draws weights $(w_1, w_2, \ldots, w_n) \sim$ Multinomial$(n; \frac{1}{n}, \ldots, \frac{1}{n})$ and computes the bootstrap statistic

$$
\begin{aligned}
\widehat{PSD}_{u,boot}^2 &= \sum_{i=1}^n \sum_{\substack{j=1 \\ i \neq j}}^n (w_i - \frac{1}{n})(w_j - \frac{1}{n})\Delta(x^{(i)}, x^{(j)}) \\
&= \sum_{k=1}^J \left(\sum_{i=1}^n (w_i - \frac{1}{n})\mathcal{A}P_k(x^{(i)})\right)^2 \\
&\quad - \sum_{k=1}^J \sum_{i=1}^n \left((w_i - \frac{1}{n})\mathcal{A}P_k(x^{(i)})\right)^2.
\end{aligned}
\tag{11}
$$

If the observed test statistic, (10), is larger than the $100(1 - \alpha)$ percentile of the bootstrap statistics, (11), then $H_0$ is rejected. The consistency of this test follows from Theorem 4.3 of Liu et al. (2016) and from Hušková & Janssen (1993). We recommend the bootstrap test because we empirically find that it has higher statistical power than the asymptotic test using samples from $Q$.

The aforementioned tests are suitable for independent samples; in the case of correlated samples, an alternate linear-time bootstrap procedure for PSD can be developed following the method of Chwialkowski et al. (2016).

## 3.3. Convergence of Moments

Since we are using a finite-dimensional (i.e. non-characteristic) kernel, PSD is not fully convergence-determining. We do not view this as a major disadvantage. Rather than aiming to detect all possible discrepancies between $P$ and $Q$ and doing so with low statistical power, the method is designed to achieve high statistical power when the discrepancy is in one of the first $r$ moments.

Often, moments of the posterior distribution, such as the mean and variance, are the main expectations of interest. These moments can also be where differences are most likely to appear; for example, the posterior variance is asymptotically over-estimated in SG-MCMC methods (Nemeth & Fearnhead, 2021). Tuning SG-MCMC amounts to selecting the right balance between a large step-size and a small step-size. Large step-sizes lead to over-estimation (bias) in the posterior variance, while small step-sizes may not sufficiently explore the space for a fixed $n$, and thus may lead to under-estimated posterior variance. Hence, it is critical to assess the performance of these methods in estimated second-order moments.

**Proposition 3.2.** *If $P$ is Gaussian with a symmetric positive-definite covariance matrix $\Sigma$, PSD $= 0$ if and only if the multi-index moments of $P$ and $Q$ match up to order $r$.*

The proof of Proposition 3.2 is provided in Appendix B. As a consequence of Proposition 3.2, we have the following result:

**Corollary 3.3.** *Suppose the conditions in Corollary 3.1 hold. Then, in the Bernstein-von Mises limit (i.e. the Bayesian big data limit), the asymptotic and bootstrap tests have power $\to 1$ for detecting discrepancies in the first $r$ moments of $P$ and $Q$ as $n \to \infty$.*

*Proof.* This result follows from the consistency of these tests and the conditions under which PSD $= 0$ as per Proposition 3.2. □

This is an important result given that biased MCMC algorithms are often used for big-data applications, with subsampling-based methods arguably being the most common application of KSD. In this context, $P$ is often close to Gaussian because typically the "Bernstein-von Mises approximation of the target posterior distribution is excellent" (Bardenet et al., 2017). We also empirically show good performance for detecting discrepancies in moments in Section 4 and in supplementary results in the Appendices.

## 4. Experiments

This section demonstrates the performance of PSD on the current benchmark examples from Liu et al. (2016), Chwialkowski et al. (2016), Jitkrittum et al. (2017) and Huggins & Mackey (2018). The proposed PSD is compared to existing methods on the basis of runtime, power in goodness-of-fit testing and performance as a sample quality measure. The following methods are the competitors:

- IMQ KSD: standard, quadratic time KSD (Gorham & Mackey, 2017; Liu et al., 2016; Chwialkowski et al., 2016) using the recommended IMQ kernel with $c = 1$ and $\beta = -0.5$.

- Gauss KSD: standard, quadratic time KSD using the common Gaussian kernel with bandwidth selected using the median heuristic.

- FSSD: The linear-time FSSD method with optimized test locations (Jitkrittum et al., 2017). We consider the optimized test locations (FSSD-opt), optimized according to a power proxy detailed in Jitkrittum et al. (2017). We set the number of test locations to 10.

- RFSD: The near-linear-time RFSD method (Huggins & Mackey, 2018). Following recommendations by

Huggins & Mackey (2018), we use the $L1$ IMQ base kernel and fix the number of features to 10.

The simulations are run using the settings and implementations provided by the respective authors, with the exception that we sample from $Q$ for all asymptotic methods since sampling from $P$ is rarely feasible in practice. Goodness-of-fit testing results for PSD in the main paper are with the bootstrap test, which we recommend in general. Results for the PSD asymptotic test with samples from $Q$ are shown in Appendix C.1. Following Jitkrittum et al. (2017) and Huggins & Mackey (2018), our bootstrap implementations for KSD and PSD use V-statistics with Rademacher resampling. The performance is similar to the bootstrap described in Liu et al. (2016) and in Section 3.2.

Code to reproduce these results is available at `https://github.com/Nars98/PSD`. This code builds on existing code (Huggins, 2018; Jitkrittum, 2019) by adding PSD as a new method. All experiments were run on a high performance computing cluster, using a single core for each individual hypothesis test.

Further empirical investigations are available in Appendix C. In Appendix C.2, we show that PSD with $r = 2$ is tracking second order moments well for the second order moment example of Kanagawa et al. (2022). We also provide two logistic regression examples in Appendices C.3 and C.4, where we demonstrate that PSD can tracks discrepancies in moments for realistic Bayesian inference tasks. The former is an example where $P$ is far from normality, while the Bernstein-von Mises approximation is reasonable for the latter. We investigate performance of PSD without interactions in Appendix C.5 and we find that the performance is remarkably similar to PSD with interactions. Finally, we provide an extreme example where moments are not well approximated and show what expectations are being tracked instead in Appendix C.6.

### 4.1. Goodness-of-Fit Tests

Following standard practice for assessing Stein goodness-of-fit tests, we begin by considering $P = N(0_d, I_d)$ and assessing the performance for a variety of $Q$ and $d$ using statistical tests with significance level $\alpha = 0.05$. We use $m = 500$ bootstrap samples to estimate the rejection threshold for the PSD and KSD tests.

We investigate four cases: (a) type I error rate: $Q = \mathcal{N}(0_d, I_d)$ (b) statistical power for misspecified variance: $Q = \mathcal{N}(0_d, \Sigma)$, where $\Sigma_{ij} = 0$ for $i \neq j$, $\Sigma_{11} = 1.7$ and $\Sigma_{ii} = 1$ for $i = 2, \ldots, d$, (c) statistical power for misspecified kurtosis: $Q = \mathcal{T}(0, 5)$, a standard multivariate student-t distribution with 5 degrees of freedom, and (d) statistical power for misspecified kurtosis: $q(x) = \prod_{t=1}^{d} \text{Lap}(x_t \mid 0, \frac{1}{\sqrt{2}})$, the product of $d$ independent Laplace distributions

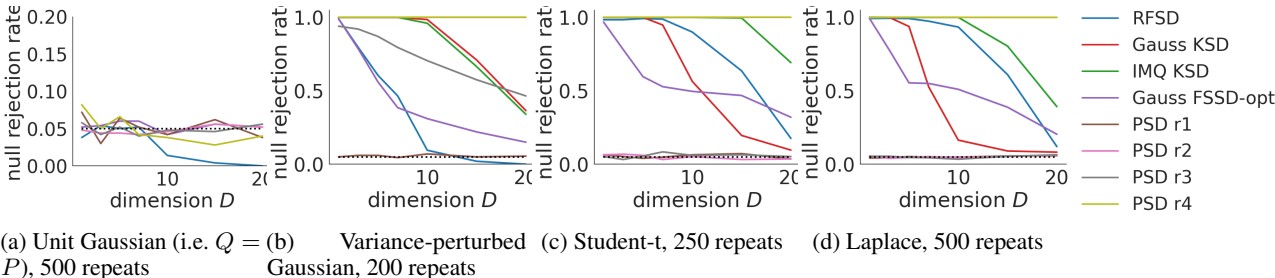

(a) Unit Gaussian (i.e. $Q = P$), 500 repeats  (b) Variance-perturbed Gaussian, 200 repeats  (c) Student-t, 250 repeats  (d) Laplace, 500 repeats

*Figure 1.* Type I error rate (a) and statistical power (b,c,d) for detecting discrepancies between the unit Gaussian $P$ and the sampling distribution $Q$ (see the main text for details).

with variance 1. Following Huggins & Mackey (2018), all experiments use $n = 1000$ except the multivariate t, which uses $n = 2000$.

As seen in Figure 1a, the type I error rate is generally close to 0.05 even with this finite $n$. Exceptions include RFSD which has decaying type I error rate with increasing $d$, and PSD with $r = 4$, which has a slightly over-inflated type I error rate for $d = 1$.

As expected, PSD with $r = 1$ is incapable of detecting discrepancies with the second (b) or fourth order moments (c,d). Similarly, PSD with $r = 2$ and $r = 3$ are incapable of detecting discrepancies with the fourth moment (c,d). When the polynomial order is at least as high as the order of the moment in which there are discrepancies, PSD outperforms linear-time methods and is competitive with quadratic-time KSD methods. Specifically, PSD with $r = 4$ is the only method to consistently achieve a power of $\approx 1$ in Figures 1c and 1d. In Figure 1b, PSD with $r = 2$ and $r = 4$ are the only methods to consistently achieve a power of 1, while PSD with $r = 3$ has a higher statistical power than the competitors at $d = 20$. Overall, the new methods have a statsistical power up to double the statistical power of IMQ KSD and four times that of linear-time competitors for $d = 20$.

Next, following Liu et al. (2016) and Jitkrittum et al. (2017), we consider the case where the target $P$ is the non-normalized density of a restricted Boltzmann machine (RBM); the samples $\hat{Q}_n$ are obtained from the same RBM perturbed by independent Gaussian noise with variance $\sigma^2$. For $\sigma^2 = 0$, $H_0 : Q = P$ holds, and for $\sigma^2 > 0$ the goal is to detect that the $n = 1000$ samples come from the perturbed RBM. Similar to the previous goodness-of-fit test, the null rejection rate for a range of perturbations using 100 repeats is given in Table 1. Notably, PSD with $r = 2$ and $r = 3$ both outperform linear-time methods and are competitive with quadratic-time methods, but for a substantially reduced computational cost. This illustrates that PSD is a potentially valuable tool for goodness-of-fit testing in the non-Gaussian setting.

*Table 1.* Null rejection rates for testing methods at different perturbation levels in the RBM example.

| PERTURBATION: | 0 | 0.02 | 0.04 | 0.06 |
|---|---|---|---|---|
| **RFSD** | 0.00 | 0.48 | 0.93 | 0.98 |
| **FSSD-opt** | 0.05 | 0.70 | 0.96 | 0.99 |
| **IMQ KSD** | 0.08 | 0.99 | 1.00 | 1.00 |
| **Gauss KSD** | 0.08 | 0.95 | 1.00 | 1.00 |
| **PSD r1** | 0.08 | 0.51 | 0.96 | 0.99 |
| **PSD r2** | 0.06 | 1.00 | 1.00 | 1.00 |
| **PSD r3** | 0.09 | 0.97 | 1.00 | 1.00 |

### 4.2. Measure of Sample Quality

To demonstrate the advantages of PSD as a measure of discrepancy we follow the stochastic gradient Langevin dynamics (SGLD) hyper-parameter selection setup from Gorham & Mackey (Section 5.3, 2015). Since no Metropolis-Hastings correction is used, SGLD with constant step size $\epsilon$ is a biased MCMC algorithm that aims to approximate the true posterior. Importantly, the stationary distribution of SGLD deviates more from the target as $\epsilon$ grows, leading to an inflated variance. However, smaller $\epsilon$ decreases the mixing speed of SGLD. Hence, an appropriate choice of $\epsilon$ is critical for accurate posterior estimation.

Similar to the experiments considered in Gorham & Mackey (2015), Chwialkowski et al. (2016) and Huggins & Mackey (2018), the target $P$ is the bimodal Gaussian mixture model posterior of Welling & Teh (2011). We compare the step size selection made by PSD to that of RFSD and IMQ KSD when $n = 10000$ samples are obtained using SGLD. Figure 2 shows the performance of SGLD for a variety of stepsizes in comparison with high quality samples obtained using MALA. Figure 3 shows that PSD with $r = 2$, $r = 3$ and $r = 4$ agree with IMQ KSD, selecting $\epsilon = 0.005$ which is visually optimal as per Figure 2. Moreover, when utilized as a measure of discrepancy, PSD is around 70 times faster than KSD and around 7 times faster than RFSD.

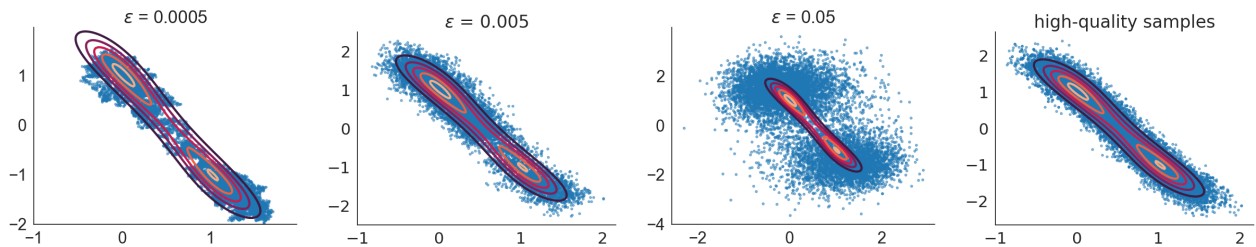

Figure 2. Approximate posterior for mixture example with SGLD for varying step sizes and when sampling from the true posterior using MALA.

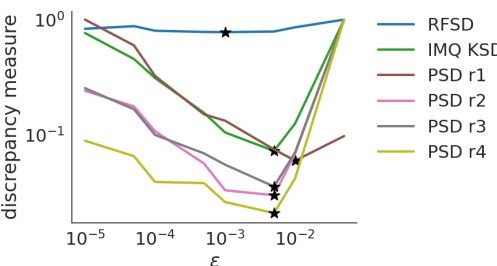

Figure 3. Step size selection results for SGLD using various methods.

### 4.3. Runtime

We now compare the computational cost of computing PSD with that of RFSD and IMQ KSD. Datasets of dimension $d = 10$ with the sample size $n$ ranging from 500 to 10000 were generated from $P = \mathcal{N}(0_d, I_d)$. As seen in Figure 4, even for moderate dataset sizes, the PSD and RFSDs are computed orders of magnitude faster than KSD. While RFSD is faster than PSD with $r = 3$ or $r = 4$, we have found in Sections 4.1 and 4.2 that PSD can have higher statistical power for detecting discrepancies in moments.

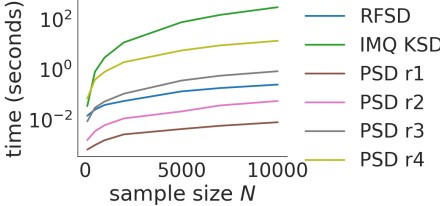

Figure 4. Runtime for various testing methods where $P = \mathcal{N}(0_d, I_d)$ with $d = 10$.

## 5. Conclusion

Our proposed PSD is a powerful measure of sample quality, particularly for detecting discrepancies in moments. The method eliminates the need for extensive tuning, is linear-time and empirically provides high statistical power in high dimensions. This makes it a valuable tool for practitioners

needing efficient, dependable measures of sample quality, especially in the context of complex Bayesian inference applications. For practitioners using PSD for biased MCMC samplers like SG-MCMC, we recommend using PSD with $r = 2$.

It is well known that probability distributions are completely determined by their moment generating functions, provided they exist (a necessary condition is that all moments are finite). This highlights the main drawback of our method, since we can potentially expect the KSD to outperform PSD for target distributions lacking well-defined moments. For example those with heavy tails such as the Cauchy distribution. However, we also note that the standard KSD (and the linear time variants) also fail for Cauchy distributions, due to Theorem 10 of Gorham & Mackey (2015). In such situations, Stein discrepancies based on diffusion Stein operators can be considered (Kanagawa et al., 2022). Similar extensions for PSD can be considered in future research

Another situation in which KSD outperforms PSD of order $r$ is when the discrepancies lie in moments higher than the $r$th order moment. For example, KSD outperforms PSD with $r = 1, 2, 3$ for the student-t and Laplace examples in Figure 1 since the discrepancy is in the kurtosis (4th order moment).

As a practical heuristic to determine whether PSD may perform well in detecting moments specifically, one could consider plotting the marginal gradients versus the marginal samples. The closer this is to a perfect linear relationship, the more one might expect the moments to be assessed. Further, if the gradients are available in analytic form, then one can also determine which expectations are being tracked by investigating the system of linear equations in equation (14) of Appendix B. We do this for the Rosenbrock target in Appendix C.6, and we are able to determine exactly which expectations are being tracked. Regardless of whether $P$ is Gaussian, the RBM, SGLD and logistic regression examples show that PSD can be a useful measure of sample quality.

The primary advantage of using the Langevin Stein operator is that, for a Gaussian target, the operator preserves the form of the monomials, and consequently we are tracking

convergence in moments. This may not hold true for other Stein operators. Nevertheless, applying the aforementioned diffusion Stein operators and other Stein operators to different types of polynomials or function classes can be studied further in future work.

We show in Appendix D that in the case of Gaussian $P$ (by extension, in the Bernstein-von Mises limit), one can apply any invertible linear transform to the parameters and the method will still detect discrepancies between the original $Q$ and $P$ in the first $r$ moments. We suggest applying whitening or simply standardization when the variances differ substantially across dimensions. Future work could also consider using alternative norms, for example by maximising subject to the constraint that $\|\beta\|_1 \leq 1$ or $\|\beta^\top W \beta\|_2 \leq 1$, the latter of which could be used to weight different monomials and therefore different moments.

Since the PSD is not translation invariant, determining the optimal parameterisation for Stein discrepancies is an open problem and an interesting point for further research. We investigate this further in Appendix C.7.

A further extension could be the use of PSD as a tool to determine the moments in which discrepancies between $Q$ and $P$ are occurring. In particular, we could examine $\tau$ and its distribution under $H_0$ to obtain an ordering of which monomial terms contribute the most to the discrepancy. Finally, theoretical investigations into the topological properties relating to the convergence of PSD can be a potential avenue for future research.

## Acknowledgments

Narayan Srinivasan is supported by the Australian Government through a research training program (RTP) stipend. LFS is supported by a Discovery Early Career Researcher Award from the Australian Research Council (DE240101190). CD is supported by a Future Fellowship from the Australian Research Council (FT210100260). LFS would like to thank Christopher Nemeth and Rista Botha for helpful discussions. Computing resources and services used in this work were provided by the High Performance Computing and Research Support Group, Queensland University of Technology, Brisbane, Australia.

## Impact Statement

This paper presents work whose goal is to advance the field of Machine Learning. There are many potential societal consequences of our work, none of which we feel must be specifically highlighted here.

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

## A. Derivation of Closed Form Solution

Consider the optimisation problem

$$
\begin{aligned}
\text{PSD} &= \sup_{g \in \mathcal{G}} |\mathbb{E}_Q[\mathcal{A}g(x)]| \\
&= \sup_{\beta \in \mathbb{R}^J : \|\beta\|_2 \le 1} \left| \mathbb{E}_Q\left[ \sum_{k=1}^{J} \beta_k \mathcal{A}P_k(x) \right] \right| \\
&= \sup_{\beta \in \mathbb{R}^J : \|\beta\|_2 \le 1} \left| \sum_{k=1}^{J} \beta_k \mathbb{E}_Q\left[ \mathcal{A}P_k(x) \right] \right| \\
&= \sup_{\beta \in \mathbb{R}^J : \|\beta\|_2 \le 1} \left| \sum_{k=1}^{J} \beta_k \bar{z}_k \right| \\
&= \sup_{\beta \in \mathbb{R}^J : \|\beta\|_2 \le 1} \sum_{k=1}^{J} \beta_k \bar{z}_k,
\end{aligned}
$$

where $\bar{z}_k = \mathbb{E}_Q[\mathcal{A}P_k(X)]$. This can be written as an optimisation problem with a Lagrange multiplier to enforce the constraint $\|\beta\|_2 \le 1$ (or equivalently $\|\beta\|_2^2 \le 1$). Specifically, we have $\text{PSD} = \sup_{\beta \in \mathbb{R}^J} L(\beta, \lambda)$, where

$$
L(\beta, \lambda) = \sum_{k=1}^{J} \beta_k \bar{z}_k - \lambda(\|\beta\|_2^2 - 1).
$$

Computing the gradient of $L(\beta, \lambda)$ with respect to $\beta$ and setting this to zero for the optimisation, we have

$$
\begin{aligned}
0 &= \nabla_\beta L(\beta, \lambda) \\
0 &= \bar{z} - 2\lambda\beta \\
\beta &= \frac{\bar{z}}{2\lambda}.
\end{aligned}
$$

Acknowledging that the supremum occurs when $\|\beta\|_2 = 1$, we have $\|\beta\|_2 = \|\frac{\bar{z}}{2\lambda}\|_2 = \frac{\|\bar{z}\|_2}{2\lambda} = 1$, so $\lambda = \frac{\|\bar{z}\|_2}{2}$ and thus $\beta = \frac{\bar{z}}{\|\bar{z}\|_2}$. Finally, substituting this solution for $\beta$ back into the original objective, the solution for the supremum is

$$
\begin{aligned}
\text{PSD} &= \sup_{\beta \in \mathbb{R}^J : \|\beta\|_2 \le 1} \sum_{k=1}^{J} \beta_k \bar{z}_k \\
&= \sum_{k=1}^{J} \frac{\bar{z}_k}{\|\bar{z}\|_2} \bar{z}_k \\
&= \|\bar{z}\|_2 \\
&= \sqrt{\sum_{k=1}^{J} \bar{z}_k^2}.
\end{aligned}
$$

## B. Proof of Proposition 3.2

*Proof.* Using the multi-index notation $x^\alpha = \prod_{i=1}^{d} x[i]^{\alpha_i}$, we have $\mathcal{G} = \text{span}\{x^\alpha : \alpha \in \mathbb{N}_0^d, \sum_{i=1}^{d} \alpha_i \le r\}$ for PSD with polynomial order $r$ and

$$
\text{PSD} = \sqrt{\sum_{\alpha \in \mathbb{N}_0^d : \sum_{i=1}^{d} \alpha_i \le r} \mathbb{E}_Q[\mathcal{A}x^\alpha]^2}.
$$

Thus, $\text{PSD} = 0$ if and only if $\mathbb{E}_Q[\mathcal{A}x^\alpha] = 0$ for all $\alpha \in \mathbb{N}_0^d : \sum_{i=1}^{d} \alpha_i \le r$. We will now proceed by proving that $\mathbb{E}_Q[\mathcal{A}x^\alpha] = 0$ for all $\alpha \in \mathbb{N}_0^d : \sum_{i=1}^{d} \alpha_i \le r$ (i.e. $\text{PSD} = 0$) *if and only* if the moments of $P$ and $Q$ match up to order $r$.

The condition that $\mathbb{E}_Q[\mathcal{A}x^\alpha] = 0$ for all $\alpha \in \mathbb{N}_0^d : \sum_{i=1}^d \alpha_i \leq r$ can be written as a system of $\binom{d+r}{d} - 1$ linear equations. For each $\alpha$, we wish to understand the conditions under which

$$
\begin{aligned}
\mathbb{E}_Q[\mathcal{A}x^\alpha] &= \mathbb{E}_Q[\Delta x^\alpha + \nabla \log p(x) \cdot \nabla x^\alpha] \\
&= \mathbb{E}_Q[\Delta x^\alpha + (-\Sigma^{-1}(x - \mu)) \cdot \nabla x^\alpha] \\
&= \mathbb{E}_Q[\Delta x^\alpha] - \mathbb{E}_Q[\Sigma^{-1}x \cdot \nabla x^\alpha] + \mathbb{E}_Q[\Sigma^{-1}\mu \cdot \nabla x^\alpha] \\
&= 0.
\end{aligned}
\tag{12}
$$

This uses the property that $P$ is a Gaussian distribution as per the Bernstein-von Mises limit, so $\nabla \log p(x) = -\Sigma^{-1}(x - \mu)$.

Since $P$ decays faster than polynomially in the tails, we can also use the property that $\mathbb{E}_P[\mathcal{A}x^\alpha] = 0$. This leads to a system of $\binom{d+k}{d} - 1$ linear equations, where for each $\alpha$

$$
\begin{aligned}
\mathbb{E}_P[\mathcal{A}x^\alpha] &= \mathbb{E}_P[\Delta x^\alpha + \nabla x^\alpha \cdot (-\Sigma^{-1}(x - \mu))] \\
&= \mathbb{E}_P[\Delta x^\alpha] - \mathbb{E}_P[\nabla x^\alpha \cdot \Sigma^{-1}x] + \mathbb{E}_P[\nabla x^\alpha \cdot \mu] \\
&= 0.
\end{aligned}
\tag{13}
$$

Subtracting (13) from (12),

$$
\mathbb{E}_{Q-P}[\Delta x^\alpha] - \mathbb{E}_{Q-P}[\nabla x^\alpha \cdot \Sigma^{-1}x] + \mathbb{E}_{Q-P}[\nabla x^\alpha \cdot \mu] = 0,
\tag{14}
$$

where $\mathbb{E}_{Q-P}[f(x)]$ is used as a shorthand for $\mathbb{E}_Q[f(x)] - \mathbb{E}_P[f(x)]$. Equation (14) combines the condition that $\mathbb{E}_Q[\mathcal{A}x^\alpha] = 0$ with the property that $\mathbb{E}_P[\mathcal{A}x^\alpha] = 0$ into a single system of equations. Our task is to prove that this system of equations holds if and only if the moments of $P$ and $Q$ match up to order $r$. We will do so using proof by induction on the polynomial order $r$.

### B.1. Base case ($r = 1$)

For $r = 1$, $x^\alpha$ simplifies to $x_i$ for $i \in \{1, \ldots, d\}$. It is simpler in this case to work directly with (12) than to work with (14). For each $i \in \{1, \ldots, d\}$, we have

$$
\mathbb{E}_Q[\mathcal{A}x_i] = 0
$$
$$
\mathbb{E}_Q[\Delta x_i] - \mathbb{E}_Q[\Sigma^{-1}x \cdot \nabla x_i] + \mathbb{E}_Q[\Sigma^{-1}\mu \cdot \nabla x_i] = 0
$$
$$
-\mathbb{E}_Q[\Sigma_{i.}^{-1}x] + \mathbb{E}_Q[\Sigma_{i.}^{-1}\mu] = 0
$$
$$
\mathbb{E}_Q[\Sigma_{i.}^{-1}x] = \Sigma_{i.}^{-1}\mu,
$$

where $\Sigma_{i.}^{-1}$ denotes the $i$th row of $\Sigma^{-1}$. Vectorising the above system of equations we have

$$
\Sigma^{-1}\mathbb{E}_Q[x] = \Sigma^{-1}\mu
$$
$$
\mathbb{E}_Q[x] = \mu.
$$

This proves that PSD with $r = 1$ is zero if and only if the first order moments (means) of $P$ and $Q$ match.

### B.2. Base case ($r = 2$)

For $r = 2$, $x^\alpha$ simplifies to $x_i x_j$ for $i, j \in \{1, \ldots, d\}$. Following (14), for each $i, j$ we have

$$
\mathbb{E}_{Q-P}[\Delta x_i x_j] - \mathbb{E}_{Q-P}[\Sigma^{-1}x \cdot \nabla x_i x_j] + \mathbb{E}_{Q-P}[\Sigma^{-1}\mu \cdot \nabla x_i x_j] = 0.
$$

The first term is $\mathbb{E}_{Q-P}[\Delta x_i x_j] = 0$ because $\Delta x_i x_j = 2\mathbb{I}_{i=j}$ does not depend on $x$ so the expectations under $P$ and $Q$ match. Similarly, the final term $\mathbb{E}_{Q-P}[\mu \cdot \nabla x_i x_j]$ disappears because first-order moments match. Thus

$$
\mathbb{E}_{Q-P}[\Sigma^{-1}x \cdot \nabla x_i x_j] = 0
$$
$$
\mathbb{E}_{Q-P}[\Sigma_{i.}^{-1}xx_j] + \mathbb{E}_{Q-P}[\Sigma_{j.}^{-1}xx_i] = 0
$$
$$
(\Sigma^{-1}\mathbb{E}_{Q-P}[xx^\top])_{ij} + (\Sigma^{-1}\mathbb{E}_{Q-P}[xx^\top])_{ji} = 0
$$
$$
(\Sigma^{-1}\mathbb{E}_{Q-P}[xx^\top])_{ij} + (\mathbb{E}_{Q-P}[xx^\top]\Sigma^{-1})_{ij} = 0
$$
$$
(\Sigma^{-1}A)_{ij} + (A\Sigma^{-1})_{ij} = 0,
\tag{15}
$$

where $A = \mathbb{E}_{Q-P}[xx^\top]$. Note that $xx^\top$ gives all possible second order moments so if $A = 0$ then all second order moments match. We need to show that $A = 0$ is the only possible solution to this equation.

Let $\Sigma^{-1} = QDQ^\top$, where $D$ is the diagonal matrix of eigenvalues with $D_{kk} = \lambda_k$ for $k \in \{1, \ldots, d\}$. This is an eigen-decomposition so $Q$ and $Q^\top$ are orthogonal ($Q^\top = Q^{-1}$). Now, (15) holds for all $i, j$ so the vectorised system of linear equations is

$$\Sigma^{-1}A + A\Sigma^{-1} = 0$$
$$QDQ^\top A + AQDQ^\top = 0$$
$$DQ^\top AQ + Q^\top AQD = 0$$
$$D\beta + \beta D = 0,$$

where $\beta = Q^\top AQ$. The third line comes from multiplying by $Q^\top$ on the left-hand side and $Q$ on the right-hand side. Consider now the $ij$th element:

$$(D\beta)_{ij} + (\beta D)_{ij} = 0$$
$$(\lambda_i + \lambda_j)\beta_{ij} = 0.$$

We know that $\lambda_k > 0$ for all $k \in \{1, \ldots, d\}$ because $\Sigma^{-1}$ is symmetric positive definite, so it must be that $\beta = 0$ and therefore $A = 0$ is the only solution.

This proves that PSD with $r = 2$ if and only if the moments of $P$ and $Q$ match up to order two, i.e. the means and (co)variances match.

### B.3. Inductive Step

Suppose that the moments up to and including order $r - 1$ match. Noting that $\mathbb{E}_{Q-P}[\Delta x^\alpha]$ is of order $r - 2$ and $\mathbb{E}_{Q-P}[\nabla x^\alpha \cdot \mu]$ is of order $r - 1$ and using (14), we have

$$\mathbb{E}_{Q-P}[\nabla x^\alpha \cdot \Sigma^{-1}x] = 0. \tag{16}$$

We must show this signifies that the moments of order $r$ must also match.

For a general $r$th order monomial, $x^\alpha$ can alternatively be written as $\prod_{m=1}^r x_{i_m}$ for $i_m \in \{1, \ldots, d\}$. Therefore we have

$$\mathbb{E}_{Q-P}[\Sigma^{-1}x \cdot \nabla \prod_{m=1}^r x_{i_m}] = \sum_{n=1}^r \mathbb{E}_{Q-P}[\Sigma_{i_n \cdot}^{-1}x \prod_{m=1, m \neq n}^r x_{i_m}] = 0.$$

This can be written in matrix form for all possible multi-indices of order $r$ using Kronecker products:

$$0 = \mathbb{E}_{Q-P}[\overbrace{\Sigma^{-1}x \otimes x^\top \otimes x^\top \otimes \cdots \otimes x^\top}^{r \text{ terms}}] + \mathbb{E}_{Q-P}[x \otimes (x^\top \Sigma^{-1}) \otimes x^\top \otimes \cdots \otimes x^\top] +$$
$$\mathbb{E}_{Q-P}[x \otimes x^\top \otimes (x^\top \Sigma^{-1}) \otimes \ldots \otimes x^\top] + \cdots + \mathbb{E}_{Q-P}[x \otimes x^\top \otimes x^\top \otimes \ldots \otimes (x^\top \Sigma^{-1})].$$

Writing $A = \mathbb{E}_{Q-P}[\overbrace{x \otimes x^\top \otimes \cdots \otimes x^\top}^{r \text{ terms}}] \in \mathbb{R}^{d \times d^{r-1}}$ and using $(A \otimes B)(C \otimes D) = AB \otimes CD$ provided that one can form the products $AB$ and $CD$, this becomes

$$\Sigma^{-1}A + A(\overbrace{\Sigma^{-1} \otimes I \otimes \cdots \otimes I}^{r-1 \text{ terms}}) + A(I \otimes \Sigma^{-1} \otimes \cdots \otimes I) + \cdots + A(I \otimes I \otimes \cdots \otimes \Sigma^{-1}) = 0.$$

Once again using $\Sigma^{-1} = QDQ^\top$, we have

$$QDQ^\top A + A(QDQ^\top \otimes I \otimes \cdots \otimes I) + A(I \otimes QDQ^\top \otimes \cdots \otimes I) + \cdots + A(I \otimes I \otimes \cdots \otimes QDQ^\top) = 0.$$

Multiplying by $Q^\top$ on the left and $(\overbrace{Q \otimes Q \otimes \cdots \otimes Q}^{r-1 \text{ terms}})$ on the right

$$0 = DQ^\top A(Q \otimes Q \cdots \otimes Q) + Q^\top A(QDQ^\top \otimes I \otimes \cdots \otimes I)(Q \otimes Q \cdots \otimes Q)$$
$$+ Q^\top A(I \otimes QDQ^\top \otimes \cdots \otimes I)(Q \otimes Q \cdots \otimes Q) + \cdots + Q^\top A(I \otimes I \otimes \cdots \otimes QDQ^\top)(Q \otimes Q \cdots \otimes Q)$$
$$0 = DQ^\top A(Q \otimes Q \cdots \otimes Q) + Q^\top A(QD \otimes Q \otimes \cdots \otimes Q)$$
$$+ Q^\top A(Q \otimes QD \otimes \cdots \otimes Q) + \cdots + Q^\top A(Q \otimes Q \otimes \cdots \otimes QD)$$
$$0 = DQ^\top A(Q \otimes Q \cdots \otimes Q) + Q^\top A(Q \otimes Q \cdots \otimes Q)(D \otimes I \otimes \cdots \otimes I)$$
$$+ Q^\top A(Q \otimes Q \cdots \otimes Q)(I \otimes D \otimes \cdots \otimes I) + \cdots + Q^\top A(Q \otimes Q \cdots \otimes Q)(I \otimes I \otimes \cdots \otimes D)$$
$$0 = D\beta + \beta(D \otimes I \otimes \cdots \otimes I) + \beta(I \otimes D \otimes \cdots \otimes I) + \cdots + \beta(I \otimes I \otimes \cdots \otimes D),$$

where $\beta = Q^\top A(Q \otimes Q \otimes \cdots \otimes Q)$. Now consider the $ij$th entries in these equations, where $i \in \{1, \ldots, d\}$ and $j \in \{1, \ldots, d^{r-1}\}$. The Kronecker product of diagonal matrices is diagonal, so the terms $(D \otimes I \otimes \cdots \otimes I)$, $(I \otimes D \otimes \cdots \otimes I)$ up to $(I \otimes I \otimes \cdots \otimes D)$ all consist of diagonal matrices with elements of $\lambda$ on the diagonal. The explicit value of $\lambda$ for any given $i$ and $j$ is not important, so for now consider indices $z_s \in \{1, \ldots, d\}$ for $s = 1, \ldots, r$ which may not be unique[2]. Then we have

$$(\lambda_{z_1} + \lambda_{z_2} + \cdots \lambda_{z_k})\beta_{ij} = 0.$$

Since all $\lambda > 0$ by positive-definiteness of $\Sigma^{-1}$, we have that $\beta_{ij} = 0$ for all $i$ and $j$ and therefore

$$\beta = 0$$
$$Q^\top A(Q \otimes Q \otimes \cdots \otimes Q) = 0$$
$$A = 0,$$

where the third line comes from multiplying by $Q$ on the left and $(Q^\top \otimes Q^\top \otimes \cdots \otimes Q^\top)$ on the right. We have now shown that $\mathbb{E}_Q[\mathcal{A}x^\alpha] = 0$ for all $\alpha \in \mathbb{N}_0^d : \sum_{i=1}^d \alpha_i \leq r$ *if and only* if the moments of $P$ and $Q$ match up to order $r$. Thus $\text{PSD} = 0$ *if and only* if the moments of $P$ and $Q$ match up to order $r$.

$\square$

# C. Additional Empirical Results

This section provides additional empirical results, as explained in Section 4.

## C.1. Asymptotic Goodness-of-fit Test

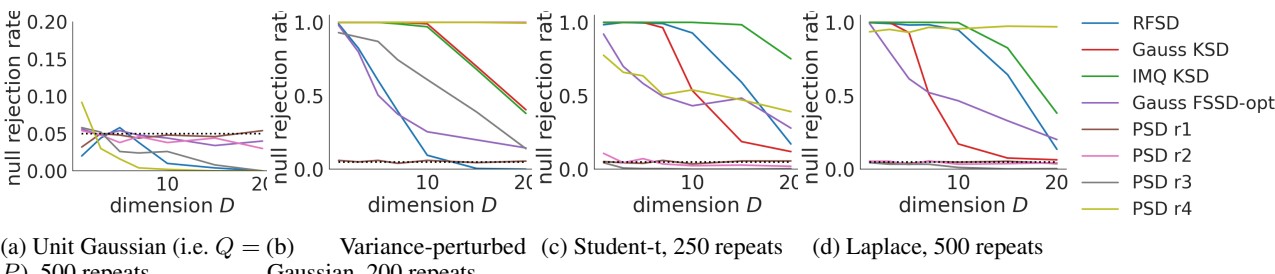

(a) Unit Gaussian (i.e. $Q = P$), 500 repeats    (b) Variance-perturbed Gaussian, 200 repeats    (c) Student-t, 250 repeats    (d) Laplace, 500 repeats

*Figure 5.* Type I error rate (a) and statistical power (b,c,d) for detecting discrepancies between the unit Gaussian $P$ and the sampling distribution $Q$, using the test based on the asymptotic distribution of the U-statistic.

We reproduce Figure 1 in Section 4.1 of the main paper by using Corollary 3.1 to implement a test based on the asymptotic distribution of the U-statistic, $\widehat{\text{PSD}}_u^2$. Specifically, we set $P = N(0_d, I_d)$ and assess the performance for a variety of $Q$

---

[2]To be explicit in an example of a fourth order polynomial, consider instead $i$ and $j$ indices starting at zero so $i \in \{0, \ldots, d-1\}$ and $j \in \{0, \ldots, d^3 - 1\}$. Then the explicit form of the equation is $(\lambda_i + \lambda_{\lfloor j/d^2 \rfloor} + \lambda_{\lfloor j/d \rfloor \% d} + \lambda_{j\%d})\beta_{ij} = 0$ where $a\%d = a - d\lfloor a/d \rfloor$ is the remainder operator.

and $d$ using statistical tests with significance level $\alpha = 0.05$. We draw 5000 samples from the null distribution given in Corollary 3.1 using the plugin estimator $\hat{\Sigma}_q$. We investigate the four cases described in Section 4.1 with identical settings as in the main paper.

The results are shown in Figure 5. Primarily, while PSD displays good performance for the variance-perturbed ($r = 2$ and $r = 4$) and Laplace ($r = 4$) experiments, PSD with $r = 4$ does not maintain high power in the case of the student-t. Also, we note that the type I error is not well controlled. Further, the asymptotic test can be slower than the bootstrap test due to the computation of the eigenvalues of the covariance matrix (complexity $\mathcal{O}(J^3)$). This can be improved by considering approximations to the covariance matrix, however this would introduce bias asymptotically.

We recommend the bootstrap procedure for goodness-of-fit testing over the asymptotic test due to its higher statistical power, better control over type I error and lower computational complexity.

## C.2. Detecting Second Moment Discrepancies

Section 4.1 of Kanagawa et al. (2022) illustrates the failure of the standard KSD with IMQ kernel for detecting non-convergence of the second moment of a simple sequence, $(Q_n)_{n \geq 1}$ to its target $P$, set to be the standard Gaussian of $d$ dimensions, $P = \mathcal{N}(\mathbf{0}, \mathbf{I}_d)$. The sequence $Q_n$ is defined as follows:

$$Q_n = \left(1 - \frac{1}{n+1}\right) P_n + \frac{1}{n+1}\delta_{x_n},$$

where $P_n = \frac{1}{n} \sum_{j=1}^{n} \delta_{X_j}$, and $x_n = \sqrt{n+1} \cdot 1$ with $1 = (1, \ldots, 1)$ (a vector of ones), and $\{X_1, \ldots, X_n\} \overset{i.i.d}{\sim} P$ are i.i.d. with $X_i \sim P$.

Kanagawa et al. (2022) demonstrate that the sequence $Q_n$ converges to $P$ almost surely. However, it has the following (almost sure) biased limit:

$$\lim_{n \to \infty} \mathbb{E}_{Y \sim Q_n}[Y \otimes Y] = \mathbb{E}_{X \sim P}[X \otimes X] + 1 \otimes 1.$$

Figure 6 shows trace plots of KSD, PSD and the Euclidean distance between the true and estimated moments up to order 2. While KSD appears to decay exponentially to zero, the trace plot for our recommended PSD with $r = 2$ is remarkably similar to the trace plot for the Euclidean discrepancy in moments up to order 2. This is in contrast to the method from Kanagawa et al. (2022), which quickly asymptotes to a non-zero value. While their approach is likely to have higher statistical power, we believe PSD has practical advantages since it is aimed at directly tracking discrepancies in moments. Moreover, we achieve moment tracking in linear time rather than the quadratic time of Kanagawa et al. (2022).

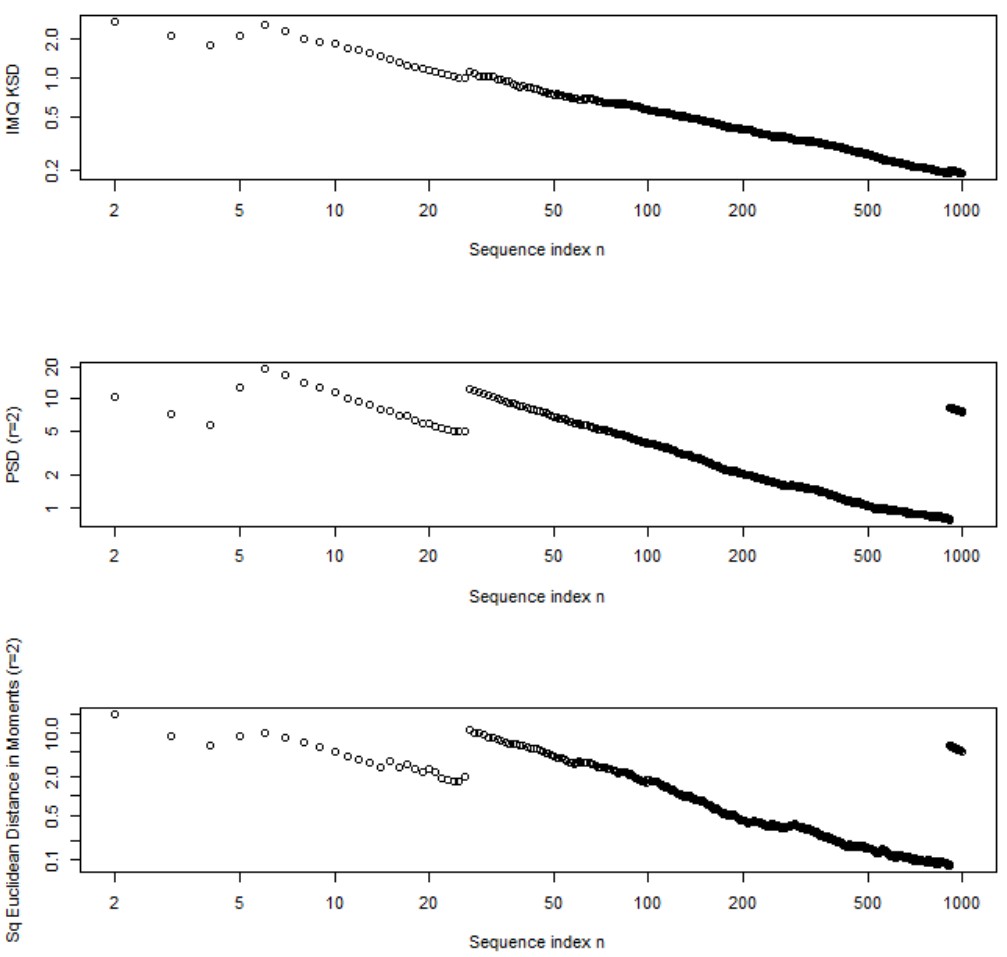

*Figure 6.* Moment tracking capabilities of PSD for Example 4.1 of Kanagawa et al. (2022)

### C.3. Logistic Regression: Sparsity Prior

We extend the empirical investigations to consider a more challenging logistic regression task with a sparsity-inducing prior. We use the stochastic search variable selection (SSVS) prior of George & McCulloch (1993), corresponding to a mixture of Gaussians with standard deviations of 0.1 and 10. The resulting posterior is multimodal with modes capturing the zero and non-zero estimates from the sparse prior. We simulate data with 100 observations and $d = 20$ variables, of which three are non-zero. We use multiple ULA runs with different step sizes and different initialisations to thoroughly investigate the performance of PSD.

Our results (averaged over 5 runs) are presented in Table 2 and the posterior distributions corresponding to the different step sizes are given in figure 7. The normalised median discrepancy values are reported. KSD and PSD with $r = 1$ have similar performance and fail to identify the poor performance of ULA for large step sizes. PSD with $r \geq 2$ identifies the step size with the best moment estimation. In particular, despite the posterior being far from normal, our methods still attain superior performance for assessing moment discrepancies compared to IMQ KSD.

| Step Size | KSD | PSD1 | Eucl1 | PSD2 | Eucl2 | PSD3 | Eucl3 | PSD4 | Eucl4 |
|---|---|---|---|---|---|---|---|---|---|
| $1 \times 10^{-5}$ | 1.0e+00 | 1.0e+00 | 6.2e-02 | 1.0e+00 | 2.2e-02 | 1.0e+00 | 7.4e-03 | 2.3e-01 | 2.4e-03 |
| $5 \times 10^{-5}$ | 5.7e-01 | 5.6e-01 | 5.8e-02 | 5.6e-01 | 2.1e-02 | 5.7e-01 | 7.2e-03 | 1.3e-01 | 2.3e-03 |
| $1 \times 10^{-4}$ | 4.3e-01 | 4.3e-01 | 5.4e-02 | 4.5e-01 | 2.0e-02 | 4.8e-01 | 6.7e-03 | 1.2e-01 | 2.2e-03 |
| $5 \times 10^{-4}$ | 2.0e-01 | 2.0e-01 | 3.9e-02 | 2.3e-01 | 1.5e-02 | 2.6e-01 | 5.0e-03 | 6.6e-02 | 1.6e-03 |
| $1 \times 10^{-3}$ | 1.4e-01 | 1.4e-01 | 3.1e-02 | 1.6e-01 | 1.2e-02 | 1.9e-01 | 3.8e-03 | 5.1e-02 | 1.2e-03 |
| $5 \times 10^{-3}$ | 7.2e-02 | 6.7e-02 | **1.56e-02** | **8.38e-02** | **4.88e-03** | **1.07e-01** | **1.79e-03** | **3.25e-02** | **5.75e-04** |
| $1 \times 10^{-2}$ | 6.6e-02 | 4.7e-02 | 7.0e-02 | 1.0e-01 | 2.9e-02 | 1.5e-01 | 1.1e-02 | 5.4e-02 | 3.8e-03 |
| $5 \times 10^{-2}$ | 2.9e-02 | 2.4e-02 | 7.6e-01 | 1.5e-01 | 6.5e-01 | 5.3e-01 | 5.4e-01 | 4.5e-01 | 4.5e-01 |
| $1 \times 10^{-1}$ | **2.78e-02** | **1.70e-02** | 1.0e+00 | 2.0e-01 | 1.0e+00 | 9.7e-01 | 1.0e+00 | 1.0e+00 | 1.0e+00 |

*Table 2.* Normalised discrepancies for the logistic regression example with sparsity-inducing prior. Shown are the KSD, the PSD and the Euclidean distance between estimated and gold-standard moments of order at most $r$. The lowest value is shown in bold.

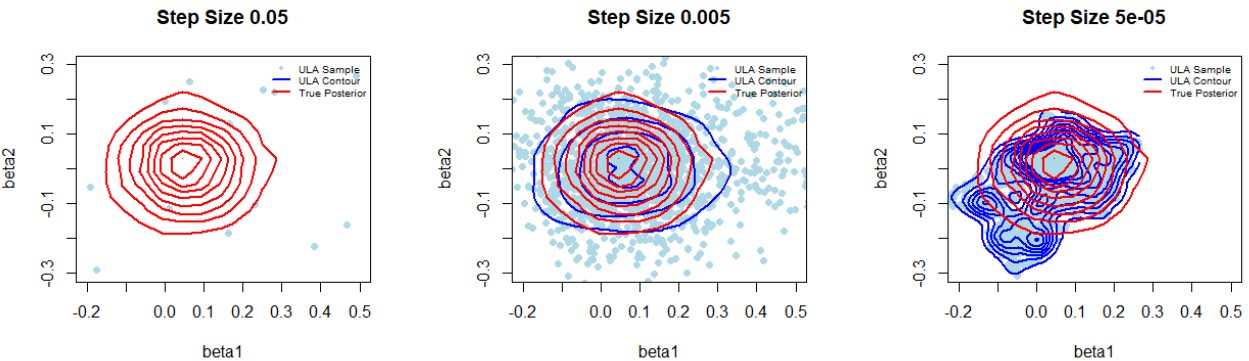

*Figure 7.* ULA sample plots in blue approximating the true posterior (red) for the logistic regression example with sparsity-inducing prior. PSD selects a step size of $0.005$ which visually provides a good approximation the true posterior.

## C.4. Logistic Regression: Big Data Example

We consider another logistic regression example looking at the performance of PSD when the posterior is close to normal. This example is a logistic regression with $10^4$ observations and $d = 5$ variables using Gaussian priors with a standard deviation of 10, thereby making the Bernstein-von Mises approximation suitable. Similarly to the previous example, we run each example five times with multiple ULA initialisations. The results are shown in Table 3 and the posterior distributions corresponding to the different step sizes given in figure 8. We find that PSD performs well in identifying the best choice of step size in ULA, though the performance for PSD with $r = 1$ is not optimal. Looking at plots of the marginal gradients versus the marginal samples as a heuristic, as per the suggestion made in Section 5, we find that when the samples are largely in the tails (e.g. for larger step-sizes), the Gaussian approximation is poor. Higher-order PSD performs excellently and in particular, PSD with $r = 2$, which is what we recommend in general, is both computationally cheap and capable of dealing with samples taken mainly in the tails.

| Step Size | PSD1 | Eucl1 | PSD2 | Eucl2 | PSD3 | Eucl3 | PSD4 | Eucl4 |
|---|---|---|---|---|---|---|---|---|
| $1 \times 10^{-5}$ | 7.44e-01 | 5.92e-05 | 1.97e-04 | 7.98e-07 | 1.80e-06 | 7.70e-09 | 1.11e-08 | 7.12e-11 |
| $5 \times 10^{-5}$ | 2.80e-01 | 2.89e-05 | 7.07e-05 | 4.22e-07 | 6.75e-07 | 4.30e-09 | 4.26e-09 | 3.70e-11 |
| $1 \times 10^{-4}$ | 2.02e-01 | 2.31e-05 | 5.14e-05 | 3.41e-07 | 4.92e-07 | 3.68e-09 | 2.76e-09 | 3.35e-11 |
| $5 \times 10^{-4}$ | 8.19e-02 | **1.40e-05** | **2.30e-05** | **2.51e-07** | **2.35e-07** | **2.97e-09** | **1.61e-09** | **2.77e-11** |
| $1 \times 10^{-3}$ | **6.25e-02** | 2.10e-05 | 3.20e-05 | 3.89e-07 | 3.74e-07 | 4.65e-09 | 2.71e-09 | 4.35e-11 |
| $5 \times 10^{-3}$ | 5.89e-01 | 2.92e-02 | 5.89e-02 | 1.19e-03 | 2.21e-03 | 3.53e-05 | 5.52e-05 | 9.13e-07 |
| $1 \times 10^{-2}$ | 7.32e-01 | 7.14e-02 | 1.19e-01 | 5.46e-03 | 7.98e-03 | 3.40e-04 | 4.71e-04 | 1.97e-05 |
| $5 \times 10^{-2}$ | 9.23e-01 | 4.60e-01 | 5.03e-01 | 2.08e-01 | 2.10e-01 | 9.19e-02 | 9.63e-02 | 4.05e-02 |
| $1 \times 10^{-1}$ | 1.00e+00 | 1.00e+00 | 1.00e+00 | 1.00e+00 | 1.00e+00 | 1.00e+00 | 1.00e+00 | 1.00e+00 |

*Table 3.* Normalised discrepancies for the logistic regression example with big data. Shown are the PSD and the Euclidean distance between estimated and gold-standard moments of order at most $r$. The lowest value is shown in bold.

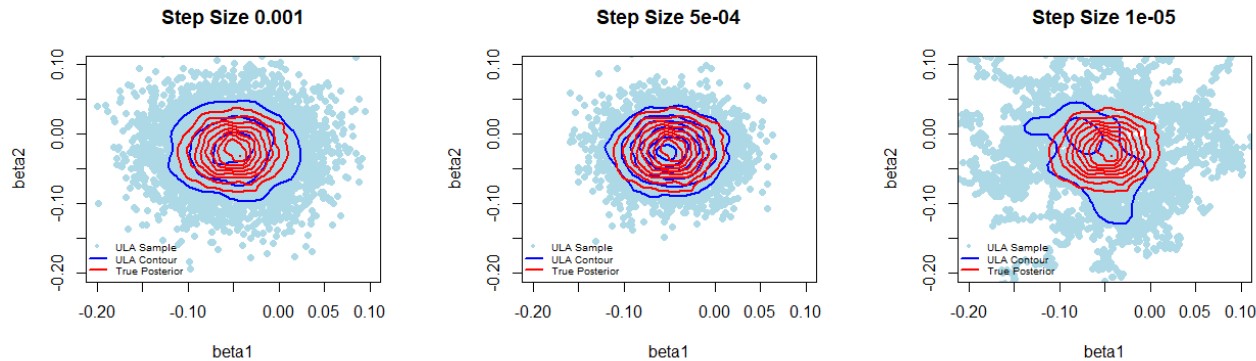

*Figure 8.* ULA sample plots in blue approximating the true posterior (red) for the logistic regression example with big data. PSD selects a step size of $5 \times 10^{-4}$ which visually provides a good approximation the true posterior.

### C.5. PSD without interaction terms

In Section 3, we proposed an alternative PSD methods that excludes interactions to achieve a computational complexity that scales linearly with dimension, specifically $\mathcal{O}(ndr)$. In the context of a Gaussian $P$ with independent components, this means the inability to detect discrepancies in correlation.

We now empirically investigate the performance of PSD without interactions on the SGLD example from Section 4.2 and on the two additional logistic regression examples. The results are shown in Figure 9 for the SGLD example and in Figure 10 for the logistic regression examples. The performance is remarkably similar to the performance with interactions. This is also true for the logistic regression examples when the covariates are simulated using an auto-regressive process with high positive or negative autocorrelation.

We do not investigate this for the Gaussian goodness-of-fit examples since we know there is no disadvantage to removing interactions in that context, and the statistical power would therefore be similar to or better than the power with interactions.

We believe PSD without interactions should generally perform similarly to PSD with interactions when applied to biased sampling methods like ULA and SGLD. Removing interactions may have more effect on bespoke sampling methods that treat multivariate relationships differently.

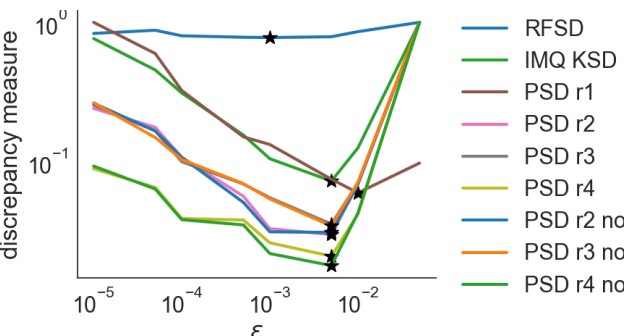

*Figure 9.* PSD with and without interactions for the SGLD example from Section 4.2.

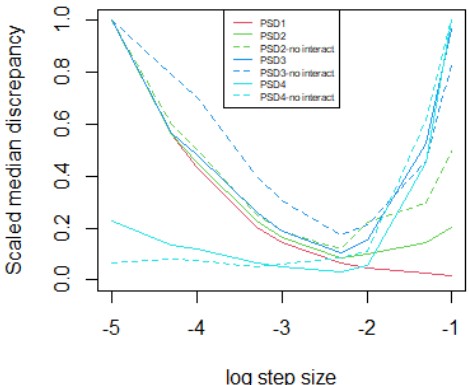
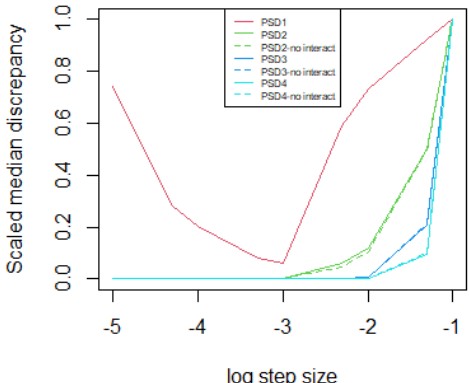

*Figure 10.* PSD with and without interactions for the logistic regression examples with (a) sparsity-inducing prior and (b) big data.

## C.6. Rosenbrock Target

We conduct an empirical study with the target $P$, set to the two-dimensional Rosenbrock function (Rosenbrock, 1960; Goodman & Weare, 2010), given by:

$$p(x) \propto \exp\left(-(x_1 - \mu_1)^2 - b\left(x_2 - x_1^2\right)^2\right),$$

where $b$ is a constant and $\mathbb{E}_P[x_1] = \mu_1$. Inference for this target is challenging because there is high probability mass in a narrow, curved region with complex dependencies and different scales across dimensions (Pagani et al., 2022).

We use this non-Gaussian example to show analytical reasoning and empirical investigations for a case where we are failing to track differences in the first $r$ moments. We provide the form of the expectations we are tracking instead when $r = 1$.

By explicitly writing the form of the PSD, we can see that PSD with $r = 1$ is zero if and only if $2b\left(\mathbb{E}_Q[x_1 x_2] - \mathbb{E}_Q[x_1^3]\right) = \mathbb{E}_Q[x_1] - \mu_1$ (associated with monomial $x_1$), and $\mathbb{E}_Q[x_1^2] = \mathbb{E}_Q[x_2]$ (associated with monomial $x_2$). This can be derived by setting $\mathbb{E}_Q[\nabla_{x_1} \log p(x)] = 0$ (associated with monomial $x_1$), and $\mathbb{E}_Q[\nabla_{x_2} \log p(x)] = 0$ (associated with monomial $x_2$) and rearranging. Unlike in PSD for Gaussian targets, this is not simply assessing the accuracy of the first moment. The equation associated with monomial $x_1$ may hold when $\mathbb{E}_Q[x_1]$ is correct (i.e. $\mathbb{E}_Q[x_1] = \mu_1$), but it could also hold when $\mathbb{E}_Q[x_1]$ is incorrect and this is offset by having $\mathbb{E}_Q[x_1 x_2] \neq \mathbb{E}_Q[x_1^3]$. Similarly, the equation associated with monomial $x_2$ is not directly assessing whether the moments of $P$ and $Q$ match, but rather whether certain relationships between moments are correct. This is different from the situation when we have Gaussian $P$.

The conditions under which PSD with higher $r$ is zero can also be derived. For example, to have zero PSD with $r = 2$ would also require that $\mathbb{E}_Q[2 + 2x_1 \nabla_{x_1} \log p(x)] = 0$ (monomial $x_1^2$), $\mathbb{E}_Q[2 + 2x_2 \nabla_{x_2} \log p(x)] = 0$ (monomial $x_2^2$) and

$\mathbb{E}_Q[x_1 \nabla_{x_2} \log p(x) + x_2 \nabla_{x_1} \log p(x)] = 0$ (monomial $x_1 x_2$). However, these become increasingly complex and difficult to interpret for higher $r$.

To empirically investigate the performance of PSD, we provide time series plots of PSD and KSD with increasing $n$ using samples either from $P$ (Figure 11) or from an altered Rosenbrock target with incorrect $b$ (Figure 12). For the former, all expectations converge to their true values as $n \to \infty$ and we expect all discrepancies to go towards zero. This is what we find empirically as well. For the latter, the first moment is correct but the variance is incorrect. One might then hope that PSD with $r = 1$ would convergence towards zero but this not the case, as we have discovered analytically and empirically in Figure 12. Importantly, since $P$ is far from Gaussian, PSD is not tracking the discrepancies in the moments well. Nevertheless, it may be a useful discrepancy, even for this unusual $P$, for certain tasks like choosing a sampler.

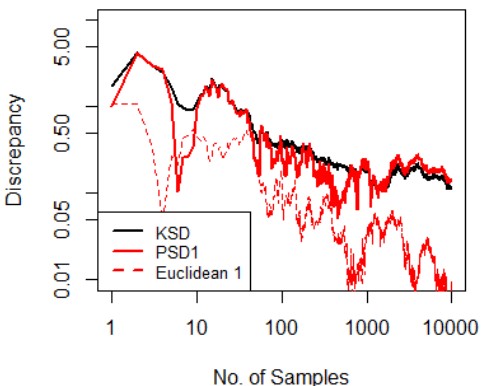
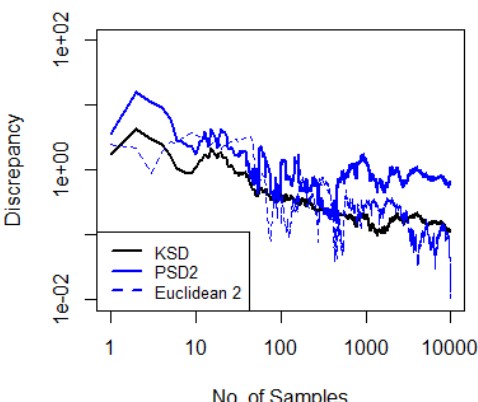

*Figure 11.* Time series plots of KSD, PSD and Euclidean distances between estimated and true moments up to order $r$ when sampling from the correct Rosenbrock target. Results are shown for (a) $r = 1$ and (b) $r = 2$.

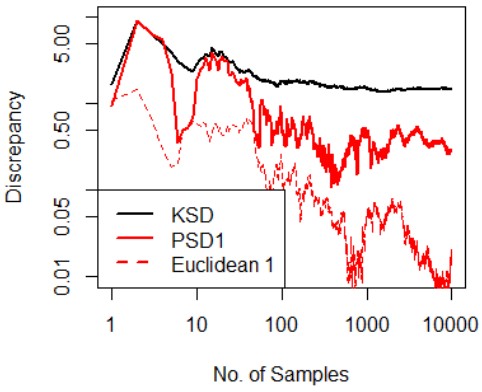
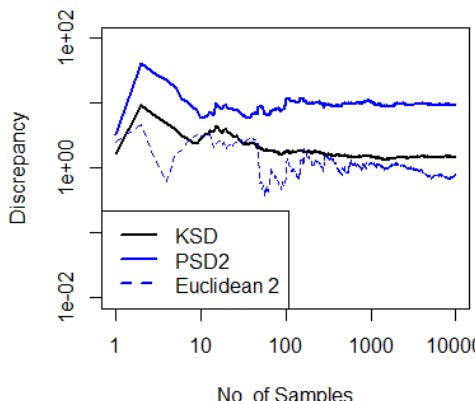

*Figure 12.* Time series plots of KSD, PSD and Euclidean distances between estimated and true moments up to order $r$ when sampling from a Rosenbrock distribution with misspecified $b$. Results are shown for (a) $r = 1$ and (b) $r = 2$.

## C.7. Translation of Mean

The PSD is not translation invariant. Specifically, one could inflate the value of PSD by considering arbitrary translations of samples by their mean. To further investigate, we have performed empirical investigations into the effect of the transformation $\tilde{x} = x - \mu_Q$, where $\mu_Q$ is the mean of $Q$. We believe this is the most sensible and practical translation to consider. However, using a mean-shift, as described, does not affect which discrepancies PSD is theoretically capable of detecting but it may affect its statistical power in doing so. It can be observed that PSD with $r = 1$ is mean-shift invariant because it is based solely on the score function, which does not change with such a transformation. However, PSD with higher $r$ uses both the samples and the score function in the discrepancy so it can be sensitive to the mean-shift.

We consider two cases based on a Gaussian $P$ with unit covariance, $N = 100$ and $d = 5$. Further, we are interested in the case where the mean of $P$ ($\mu_P$) is not the same across all dimensions, since we believe this is where the impact on statistical power will have the most effect. For this reason, we consider two cases, (1) $\mu_P = (a, 0, \ldots, 0)$ and (2) $\mu_P = (0, a, \ldots, a)$. The discrepancy will be in the first dimension in both cases, so we are interested in how the relative scales of the means that are correctly specified (versus misspecified) affect the results. We consider the two cases of mean translation for situations with a misspecified mean as well as misspecified variance.

Table 4. We consider both case (1) and case (2) in situations where the discrepancy is in the (first) mean ($\mu_Q = \mu_P + 0.5e_1$) or in the (first) variance ($[\Sigma_Q]_{11} = [\Sigma_P]_{11} + 0.5$). A "t" in front of the discrepancy indicates we have performed a mean-shift reparameterisation.

| Case | Misspecified | Discrepancy | -1.5 | -1.0 | -0.5 | 0.0 | 0.5 |
|------|--------------|-------------|------|------|------|-----|-----|
| 1 | Mean | $PSD_2$ | 0.70 | 0.11 | 0.06 | 0.16 | 0.87 |
| | | $tPSD_2$ | 0.98 | 0.22 | 0.06 | 0.13 | 0.84 |
| | | $PSD_3$ | 0.71 | 0.26 | 0.06 | 0.18 | 0.76 |
| | | $tPSD_3$ | 0.92 | 0.30 | 0.05 | 0.19 | 0.74 |
| 2 | Mean | $PSD_2$ | 1.00 | 1.00 | 0.76 | 0.20 | 0.08 |
| | | $tPSD_2$ | 1.00 | 1.00 | 0.84 | 0.14 | 0.06 |
| | | $PSD_3$ | 1.00 | 0.96 | 0.72 | 0.26 | 0.06 |
| | | $tPSD_3$ | 1.00 | 1.00 | 0.82 | 0.25 | 0.10 |
| 1 | Variance | $PSD_2$ | 1.00 | 0.96 | 0.78 | 0.55 | 0.32 |
| | | $tPSD_2$ | 1.00 | 0.98 | 0.84 | 0.54 | 0.32 |
| | | $PSD_3$ | 0.96 | 0.92 | 0.70 | 0.36 | 0.17 |
| | | $tPSD_3$ | 0.97 | 0.90 | 0.70 | 0.32 | 0.17 |
| 2 | Variance | $PSD_2$ | 0.97 | 0.83 | 0.62 | 0.39 | 0.27 |
| | | $tPSD_2$ | 1.00 | 0.98 | 0.88 | 0.56 | 0.34 |
| | | $PSD_3$ | 0.83 | 0.65 | 0.44 | 0.29 | 0.16 |
| | | $tPSD_3$ | 0.98 | 0.90 | 0.70 | 0.36 | 0.16 |

Table 4 shows the estimated statistical power based on 200 independent simulations. The results demonstrate that the original mean scaling does affect the results, but the performance with the mean-shift reparamerisation is generally similar to, if not slightly better than, the performance with no reparameterisation. Importantly, the choice of parameterisation is a problem that affects Stein discrepancies more generally. KSD with radial kernels, i.e. kernels that are functions of the form $\|\|x - y\|\|$, like the Gaussian kernel, are mean-shift invariant. However, they are still sensitive to other types of reparameterization, such as whitening (considered in Appendix D).

# D. Using an Invertible Linear Transform

**Corollary D.1.** *Consider PSD applied on the transformed space $y = Wx$, where $W \in \mathbb{R}^{d \times d}$ is an invertible matrix that is independent of $\{x_i\}_{i=1}^N$. Denote the distribution of $y = Wx$ by $\tilde{Q}$. Using a change of variables, $\nabla_y \log \tilde{p}(y) = W^{-\top} \nabla_x \log p(x)$. The new discrepancy,*

$$PSD_W = \sqrt{\sum_{\alpha \in \mathbb{N}_0^d : \sum_{i=1}^d \alpha_i \leq r} \mathbb{E}_{\tilde{Q}}[\Delta_y y^\alpha + \nabla_y \log \tilde{p}(y) \cdot \nabla_y y^\alpha]^2},$$

*is zero if and only if the moments of $P$ and $Q$ match up to order $r$.*

*Proof.* Under the Bernstein-von-Mises limit (or generally for Gaussian targets), $P = \mathcal{N}(\mu, \Sigma)$ so in the transformed space $\tilde{P} = \mathcal{N}(\tilde{\mu}, \tilde{\Sigma})$ where $\tilde{\mu} = W\mu$ and $\tilde{\Sigma} = W\Sigma W^\top$.

Next, we will show that $\tilde{\Sigma}$ is symmetric positive definite. Since $\tilde{\Sigma} = W\Sigma W^\top = \tilde{\Sigma}^\top$, $\tilde{\Sigma}$ is symmetric. Since $\Sigma$ is positive-definite, $z^\top \tilde{\Sigma} z = z^\top W\Sigma W^\top z > 0$ only requires that we do not have $W^\top z = 0$ and therefore by invertibility of $W$, that we do not have $z = 0$. Thus, $z^\top \tilde{\Sigma} z = z^\top W\Sigma W^\top z > 0$ for all non-zero $z \in \mathbb{R}^d$, which by definition means that $\tilde{\Sigma}$ is positive-definite.

The assumptions of Proposition 3.2 (symmetric, positive-definite covariance) are met for this transformed space so for any $r \in \mathbb{N}$ we have

$$
\mathbb{E}_{y \sim \tilde{Q}}[\overbrace{y \otimes y^\top \otimes \cdots \otimes y^\top}^{(r \text{ terms})}] = \mathbb{E}_{y \sim \tilde{P}}[\overbrace{y \otimes y^\top \otimes \cdots \otimes y^\top}^{(r \text{ terms})}]
$$

$$
\mathbb{E}_{x \sim Q}[Wx \otimes (Wx)^\top \otimes \cdots \otimes (Wx)^\top] = \mathbb{E}_{x \sim P}[Wx \otimes (Wx)^\top \otimes \cdots \otimes (Wx)^\top]
$$

$$
W\mathbb{E}_{x \sim Q}[x \otimes x^\top \otimes \cdots \otimes x^\top](W^\top \otimes \cdots \otimes W^\top) = W\mathbb{E}_{x \sim P}[x \otimes x^\top \otimes \cdots \otimes x^\top](W^\top \otimes \cdots \otimes W^\top)
$$

$$
\mathbb{E}_{x \sim Q}[x \otimes x^\top \otimes \cdots \otimes x^\top] = \mathbb{E}_{x \sim P}[x \otimes x^\top \otimes \cdots \otimes x^\top].
$$

Therefore we have that PSD applied to the transformed $y = Wx$ is equal to zero if and only if the moments of $P$ and $Q$ match up to order $r$.

$\square$

