# OpenReview forum: "The Polynomial Stein Discrepancy for Assessing Moment Convergence"
_ICML.cc/2025/Conference — ICML 2025 poster_

### Official Review · Reviewer_JxPo · 2025-03-07

**Overall Recommendation:** 3

**Summary:**

In Bayesian statistics, it is quite common to want to integrate some
functions with respect to the posterior distribution (e.g. the mean). When
the posterior is complicated, this has no closed form solution, and so
practitioners often resort to using MCMC samplers or diffusion samplers. In
the past decade, the Stein discrepancy was developed as a way to quantify
the quality of a sample to a target distribution, even if the
normalization constant of the target distribution was unknown. The typical
choice of the Stein set is usually an RKHS associated with the Gaussian or
inverse multiquadric kernel.

The authors propose using a different Stein set here, i.e., a Stein set
where the preimage is the vector space of all monomials with degree less
than or equal to $r$. The authors argue the benefits are twofold: this can
be more computationally efficient when $r$ and $d$ are relatively small
compared to $n$ and in the case of the target is Gaussian, it focuses the
discrepancy more on the similarity of the moments of the two
distributions. They offer some theory showing that this holds in the of a
Gaussian and they also illustrate the efficacy of this discrepancy on a many
different examples commonly used in this literature.

## update after rebuttal
Based on the authors feedback, I'm inclined to keep my score of a weak accept. I think there are good ideas in the paper, but there are still some lingering questions (choice of polynomial basis, lack of shift invariance, etc) which degrade the practicality of the method.

**Claims And Evidence:**

Yes, the claims in this paper are convincing. The main theoretical claim
(Proposition 3.2) is proved in the appendix, and all other claims about its
efficacy are demonstrated empirically. This is ample empirical work, which
both displays the benefits of this approach, and also some areas where the
Gaussian assumption of $P$ can go wrong.

**Essential References Not Discussed:**

The paper did a good job citing the relevant literature.

One thing I might suggest is that in "Measuring sample quality with Stein’s method" paper, the graph Stein discrepancy actually does control convergence with respect to the Wasserstein metric, which implies convergence in mean.

**Experimental Designs Or Analyses:**

The experimental analysis here seem valid. Most of them are borrowed from
previous papers in the field and thus have been used as benchmarks before.

**Methods And Evaluation Criteria:**

Yes, the proposed methods make intuitive sense for the problem at
hand. The authors even show that the PSD can perform well for distributions
that are bimodal (Figure 2), but that it can also fail for some non-Gaussian
and heavy tailed targets (Appendix C.6).

They also demonstrate both theoretically and empirically that for low $r$,
this method is computationally more efficient than alternatives.

**Other Comments Or Suggestions:**

+ A few places in the paper mention "Theorem 3.1" but in the paper this is
  written as "Corollary 3.1."

+ It would be nice if Figure 4 was a log/log plot so it was easier to see the
different growth rates.

**Other Strengths And Weaknesses:**

The main strength of this paper is its signficance in articulating an
efficient discrepancy that can help identify differences in moments when the
target distribution is approximately Gaussian. The idea of using a Stein set
of monomials is not so novel, but the thorough empirical work and ability to
work in a broad range of examples is quite remarkable.

The simplicity of the method and clarity of the paper are also positives.

There are some limitations to the paper. In the general theory of Stein discrepancies,
one often defines the suitable test set $\mathcal{H}$ which controls convergence.
E.g., in another version of the Stein discrepancy, this would be the set currently used
as the span of monomials $\mathcal{G}$. Then one usually solves the Stein equation
$\mathcal{A} g  = h$ so they can guarantee there is always a $g\in\mathcal{G}$ that
satisfies the Stein equation for each $h$ for some set $\mathcal{G}$.

The authors here do implicitly show a version of this for multivariate Gaussian $P$, which is a notable feat. However,
as discussed in the questions below, this does not naturally generalize to non-Gaussian
distributions. The authors do present some solid evidence that the PSD can still perform
well even for non-Gaussian $P$, but there are no robustness results to illustrate when
the PSD might behave quite surprisingly. And while the point about the Bernstein-von Moses
limit is well taken, it still doesn't provide much confidence how powerful this statistic is when
using the PSD in the wild.

**Questions For Authors:**

[Q1] Under what conditions should a practitioner elect to use the PSD over the
KSD (with say IMQ kernel)? As surfaced in the paper, the IMQ-KSD at least controls
weak convergence (but not necessarily moments), whereas the PSD controls moments in the case when $P$ is Gaussian.
If practitioners know that the posterior distribution is Gaussian, they can surely use
the PSD, but how often do practitioners know this in advance? If practitioners know
that the target is Gaussian, aren't there simpler ways to learn the moments of the
target distribution? If the moment matching were guaranteed for a larger class of
target distributions (e.g. log concave), then perhaps the validity of this would be
more concrete.

[Q2] The PSD given here is not translation invariant, i.e., if one shifts
both P and Q by some vector $\alpha$, the PSD will have a different
value. It appears that the implicit choice of the origin imposes different
weights on each basis monomials of $\mathcal{G}$. How sensitive is the power of the
PSD to this implicit parameterization? Should one try to center the
distributions $P$ and $Q$ before using the PSD?

[Q3] The authors show that if $P$ is a multivariate Gaussian and $PSD(P, Q)=0$,
then the first $r$ moments of $P$ and $Q$ match. Can the authors say anything
about the topology of the PSD, even in the case that $P$ is Gaussian? I.e.,
if $PSD(P, Q_n)\to 0$, does this imply that the first r moments of $Q_n$ must converge
to $P$?

[Q4] The authors mention that the PSD presented here is "offering a simpler formulation
that may be more effective for identifying specific moments where discrepancies occur."
Can the authors explain this in more detail? In the non-Gaussian case this seems
non-trivial, and even in the Gaussian case, how would this look exactly? The differentiation operator and a non-identity covariance
would result in some weighted sum of monomials, which are not exactly known without knowing the covariance. Are there any
benefits to using a different set of spanning polynomials, e.g. the Chebyshev polynomials?

**Relation To Broader Scientific Literature:**

The paper is motivated by tailoring the Stein discrepancy to be
even more useful than it currently is for Bayesian analysis. Since often
practitioners often want their MCMC samplers to approximate a Bayesian
posterior in the mean and covariance, this discrepancy is more tailored to
helping identify when that has gone awry.

**Theoretical Claims:**

The authors show that PSD = 0 enforces moment matching up to order $r$ when
$P$ is Gaussian. This illustrates that the PSD does at least what
practitioners want in the most simple case. This was fully argued in
Appendix B.

---

> ### Author Rebuttal · Authors · 2025-04-01
>
> Thank you for your time reviewing our manuscript and for your valuable comments and suggestions. We will create a log/log plot, replace "Theorem 3.1" with "Corollary 3.1." and mention that property of the graph Stein discrepancy.
>
> **(Lack of) confidence about power of PSD in the wild**
>
> We have highlighted in the paper the strong theoretical performance on Gaussian targets and the strong empirical performance of PSD on non-Gaussian targets. During this review period, we have also highlighted issues with PSD for distributions where moments do not exist (e.g. Cauchy).  However, the development of theoretical results for the non-Gaussian setting represents a substantial undertaking that would be better considered in a future paper.
>
> **When to use PSD over KSD**
>
> Broadly we suggest using PSD when one wishes to obtain high statistical power for detecting discrepancies in moments and the target is "close to Gaussian" (with checks that we have expanded on elsewhere). Fast but biased sampling methods like SG-MCMC, where the discrepancy is often in the variance and the target is often close to Gaussian, are good applications that we highlight in the paper. The linear-time complexity of PSD is particularly helpful for large $n$.
>
> **How do practitioners know the posterior is Gaussian?**
>
> We propose a diagnostic plot in the discussion. If the gradients are available in analytic form, rather than indirectly through automatic differentiation procedures, then one can also determine which expectations are being tracked by analysing the system of linear equations in equation (14) of Appendix B. We do this for the Rosenbrock target in Appendix C.6, and we are able to determine exactly which expectations are being tracked. We will suggest this technique in the discussion to diagnose situations where the PSD might not behave as expected.
>
> **Simpler ways to learn the moments if the target is Gaussian**
>
> Estimating the first $r$ moments need not be our only goal. However, checking for discrepancies in moments can be important in practice. For example, in the common application of SG-MCMC algorithms, poor step-sizes can lead to under- or over-estimated variance. Selecting the step-size based on matching the second-order moment can lead to a reasonable posterior approximation, as explained in the paper.
>
> **Moment-matching guarantees for a larger class of target distributions**
>
> We can guarantee PSD is assessing moments for Gaussian targets because $\nabla \log p(x)$ is a linear function in that context, so the Stein operator applied to an $r$th order polynomial is itself an $r$th order polynomial. In this sense, the closer $\nabla \log p(x)$ is to a linear function, the closer the PSD is to assessing discrepancies in the first moments.
>
> **Sensitivity of PSD to mean-shift**
>
> We are investigating this further by observing the sensitivity of PSD to transformations. The moment-tracking property of PSD will continue to hold due to similar reasoning as Appendix D.
>
> **Topology of PSD and properties when $PSD(P, Q_n)\to 0$**
>
> While higher order PSD becomes increasingly complex, we can explicitly consider the case of $P=N(\mu_P, \Sigma_P)$ and $r = 1$. Here, $PSD(P,Q_n) = ||n^{-1}\sum_{i=1}^n\Sigma_P^{-1}(\mu_P-X_i) ||_2 = ||\Sigma_P^{-1}(\mu_P-\bar{X}) ||_2$ is measuring the Mahalanobis distance between $\mu_P$ and the sample mean $\bar{X}$. From the law of large numbers as $n\rightarrow \infty$, $\bar{X}\rightarrow \mu_Q$ and consequently, $PSD(P,Q)\rightarrow  \Sigma_P^{-1}(\mu_P-\mu_Q)$ is zero if and only if the moments of $P$ and $Q$ match. Assuming iid draws from $Q$, the CLT gives $\sqrt{n}\Sigma_P^{-1}(\bar{X}_Q-\mu_q)\sim N(0, \Sigma_P^{-1}\Sigma_Q\Sigma_P^{-T})$ so $PSD = ||Y +\Sigma_P^{-1}(\mu_q-\mu_P)||_2$ where $Y\sim N(0, n^{-1}\Sigma_P^{-1}\Sigma_Q\Sigma_p^{-T})$. The PSD behaves as the square root of a noncentral $\chi^2$ distribution.
>
> This result for $r=1$ serves as an illustrative example for iid samples and Gaussian targets. More practically useful theory, including for correlated samples, would be substantially more challenging and we will mention it as future work in the paper.
>
> **Simpler formulation for identifying specific moments where discrepancies occur**
>
> If the target is Gaussian with diagonal covariance, then for the form of PSD we consider, the individual terms in the PSD correspond exactly to the moments of interest, as explained in the paper. This makes identification of problematic moments simpler than for alternative polynomial kernels.
>
> **Benefits of other spanning polynomials?**
>
> Unfortunately the benefits of orthogonal polynomials such as Chebyshev polynomials would likely be lost with PSD. It is not clear that one obtains a Chebyshev polynomial after applying a Langevin Stein operator to a Chebyshev polynomial.

---

### Official Review · Reviewer_8qrD · 2025-03-09

**Overall Recommendation:** 4

**Summary:**

This paper proposes a class of monomials as test functions in kernelised Stein discrepancies, in order to speed up computations. It shows that when the target is Gaussian then the method works well.

**Claims And Evidence:**

The claims are

* the method detects differences in the first r moments in the Bernstein-von-Mises limit;
* the corresponding KSD test has higher power than some of its competitors, in some empirical examples.

It is the first claim that I find misleading. The paper shows that when the target is Gaussian, the method detects differences in the first r moments. It somewhat artificially puts the result in a Bayesian framework by instead of clearly stating that the method works well on Gaussian target, the Gaussian is called the `Bernstein-von-Mises limit'.

**Essential References Not Discussed:**

None beyond the above mentioned papers.

**Experimental Designs Or Analyses:**

The experimental setup in the main paper is standard and useful to compare against other distributions. In the supplementary material also a Rosenbrock target is used. I found it difficult to get more information about this distribution; it seems to have an unknown normalising constant and a narrow ridge around the mode. This target distribution is clearly not Gaussian and the simulations show that the proposed method does not work very well in this case.

**Methods And Evaluation Criteria:**

The empirical evaluations look ok, although the code is not available (for blinded review, it states, but it is possible to put the code on an anonymous github site).

**Other Comments Or Suggestions:**

Whether or not the rth moments exist is not discussed in he paper.

**Other Strengths And Weaknesses:**

The idea of using polynomials as test functions is new in the context of KSD. However in my view the paper is hiding the key results under the notion of Bernstein-von-Mises limit, when really it means that the target is Gaussian. This could be seen as misleading.

**Questions For Authors:**

* What happens if the target does not have any moments?

* How much do the results depend on the Langevin Stein operator? What happens if a different Stein operator is used?

**Relation To Broader Scientific Literature:**

In the area of Stein's method there are results available for multivariate normal approximation using as test functions those of polynomial growth,

Gaunt, Robert E. "Stein’s method for functions of multivariate normal random variables." (2020): 1484-1513.

It would be good to relate the approach in the paper to theoretical underpinnings; in particular bounds on convergence rates could be of interest.

It would also be good to relate the results to quantitative Bernstein-von-Mises theorems such as the ones mentioned above.

The resampling method is very related to

Xu, Wenkai, and Gesine D. Reinert. "A kernelised Stein statistic for assessing implicit generative models." Advances in Neural Information Processing Systems 35 (2022): 7277-7289.

That paper includes normal approximation as a special case and gives explicit theoretical guarantees for re-sampling.

**Theoretical Claims:**

The proofs are correct from what I see.

However the method is based on comparing high moments. If the underlying distribution is, say, Cauchy, and not Gaussian, then it is not clear that the method would work at all.

While the limiting behaviour of posterior mean (or mode or median) often follows a Gaussian distribution, this is what I believe the authors call the Bernstein-von-Mises limit. This convergence only holds under some conditions; see for example papers by Spokoiny et al. and Kasprzak et al.  which show that the rate of convergence depends on the moments. Hence when the moments in the test are high, the posterior mean may still be far from normal and hence the asympotic regime is not warranted.

---

> ### Author Rebuttal · Authors · 2025-04-01
>
> Thank you for your time reviewing our manuscript and for your valuable comments and suggestions.
>
> **On the BvM limit**
>
> Proposition 3.2 explicitly states the Gaussianity assumption without mentioning the BvM limit, but we agree that this can be clarified in other parts of the text. We will reword the text in the abstract and the introduction of the paper. We will replace the text "we prove that it detects differences in the first $r$ moments in the Bernstein-von Mises limit" in the abstract with "we prove that it detects differences in the first $r$ moments for Gaussian targets."
>
> We opt to keep the text about the BvM as a remark after our theoretical result. This is relevant to the practical application of the method because a major application of KSD and related methods is in fast but biased Monte Carlo methods, such as SG-MCMC, where posterior distributions are often close to Gaussian. Bardenet et al (2017) highlight this in their review paper on MCMC for tall data, stating ``we emphasize that we have only been able so far to propose subsampling-based methods which display good performance in scenarios where the Bernstein-von Mises approximation of the target posterior distribution is excellent. It remains an open challenge to develop such methods in scenarios where the Bernstein-von Mises approximation is poor."
>
> **Method is based on comparing high moments**
>
> As mentioned in Section 3.3, lower-order moments are often of interest and they can be where discrepancies are most likely to appear. We give the example of SG-MCMC, where large step-sizes lead to over-estimation of the posterior variance and small step-sizes can lead to under-estimation of the posterior variance. We therefore do not necessarily agree that the method is based on comparing high moments.
>
> **Rosenbrock target context**
>
> We will explain that it is challenging because there is high probability mass in a narrow, curved region with complex dependencies and different scales across dimensions. We will add references to the original work (Rosenbrock, 1960), the probabilistic version (Goodman and Weare, 2010) and a paper talking about its complexity (Pagani, 2022).
>
> **Gaunt (2020) and relations to theoretical underpinnings**
>
> Thank you for the reference. Since the polynomial functions are not Lipschitz, we cannot directly compute the solution of Stein's equation and establish bounds on the derivative. The suggested paper seems to describe a technique to establish the required bounds for unbounded test functions which seems suitable for the PSD case. However, it is not clear how to establish this for our formulation of PSD.  Further, the main goal of the paper seems to focus on establishing the closeness of the distribution of $g(W)$ to the distribution of $g(Z)$, where $W$ is a sequence of random variables approximating a standard Gaussian. We are unsure of the relevance of this continuous mapping theorem to our problem - what would be a possible interpretation of the function $g$?
>
> Initial results on the convergence of PSD for a simple setting are given in response to Reviewer JxPo.
>
> **Relationship to Xu et al (2022)**
>
> The resampling method is indeed similar to the method we have considered. Following Liu et al (2016), we are able to establish the asymptotic exactness of our bootstrap procedure. Theoretical guarantees for resampling under a normal approximation could be a valuable reference to extend the method. Upon our preliminary review of that paper, we feel that the results in Appendix $B.1$ are of interest and we will cite this as a potential extension for the bootstrap procedure. Please let us know if you were referencing a different theorem or section of the paper.
>
> **What if moments do not exist?**
>
> We believe this could affect things in two ways. First, we might not have the critical zero expectation property for the Stein operator applied to the polynomial, hence we might not expect that PSD $= 0$ when $P = Q$. Second, we will not be assessing moments exactly because the gradients of the log target will not be linear in $x$. We will mention limitations around moments not existing in the discussion.
>
> **Dependence on Langevin Stein operator**
>
> The advantage of using the Langevin Stein operator is that when it is applied to monomials for a Gaussian target then we obtain monomials back, and therefore we are tracking convergence in moments. This may not hold true for other Stein operators. Nevertheless, applying diffusion Stein operators and other Stein operators, such as those considered in Kanagawa et al (2022), to different types of polynomials is an interesting question and can be studied further in future work. We will mention this in the discussion.
>
> **Anonymous Github site**
>
> The full set of code has been provided as a zipped folder in the supplementary material for (optional) review.  We would prefer to use the first author's Github account if the paper is accepted to encourage appropriate attribution.

---

> > ### Comment · Reviewer_8qrD · 2025-04-03
> >
> > Thank you for the explanations. As you will take the comments and suggestions on board I am happy to adapt my recommendation to accept.

---

> > > ### Author Response · Authors · 2025-04-09
> > >
> > > Thank you for your comments and for adapting your recommendation to accept. We would like to follow up our earlier response about the case where moments do not exist with information about the performance of KSD for the Cauchy distribution. Theorem 10 from Gorham and Mackey (2015) shows that KSD with standard bounded kernels ($C_0$) fails to dominate weak convergence when the target has a bounded score function. In the case of the Cauchy distribution, $\mid \nabla \log p(x) \mid = \mid \frac{2x}{1+x^2} \mid \leq 1$. Hence it follows that KSD (and the linear time variants) also fail for Cauchy distributions. We will add this to our discussion about applications where moments do not exist.

---

### Official Review · Reviewer_7ab6 · 2025-03-14

**Overall Recommendation:** 4

**Summary:**

The paper proposes a Stein discrepancy metric (PSD) for hypothesis testing. Rather than attempt to target a broad set of functions, the paper argues that by restricting to polynomial moments (up to some order), one can obtain a metric that is light-weight for computation and also has better statistical power.

**Claims And Evidence:**

The authors claim some empirical benefits and provide appropriate benchmarks. They also provide a method for approximating their discrepancy metric from samples, and compute some asymptotics to characterize the quality of the approximation.

**Essential References Not Discussed:**

As far as I am aware, the most relevant references have been covered.

**Experimental Designs Or Analyses:**

The paper provides some assessments of PSD against other common KSDs, for common/fairly standard data sets.

**Methods And Evaluation Criteria:**

Yes, the benchmarks are thorough and the comparison between methods seems fair.

**Other Comments Or Suggestions:**

Line 322: sfix -> fix

**Other Strengths And Weaknesses:**

I find the main idea behind this work to be rather compelling; namely, I like this concept of testing against a smaller subset of functions. I believe this could be interesting if developed further, for instance by considering other classes of test functions.

My main concern is that the paper could have benefited from further exploration of some of the ideas contained therein.

Nonetheless, the statistic is simple, computationally cheap, and I believe would certainly be of use to practitioners. Thus, I would recommend that this paper be accepted as it stands.

**Questions For Authors:**

What if I wanted to test against a specific set of functions other than those given in the paper? Is there any other example of this generating an interesting set of statistics? For instance, if I wanted to test against the moments after applying some transform to my random variable, could I possibly concoct interesting tests this way?

**Relation To Broader Scientific Literature:**

The paper relates to known literature on the construction/selection of KSDs. More specifically, it discusses common KSDs in the context of their computational tractability. The relative merits and downsides of various prior schemes are mentioned, in order to highlight the need for KSDs which are easily computed and yet which still have statistical advantages.

**Theoretical Claims:**

As noted in claims, the primary theoretical claim relates to the asymptotics of their approximate statistics. There are also detailed derivations to justify correctness of discrepancy metric.

---

> ### Author Rebuttal · Authors · 2025-04-01
>
> Thank you for your time reviewing our manuscript and for your valuable comments and suggestions. We will correct the typographical error.
>
> **What if I wanted to test against a specific set of functions other than those given in the paper? Is there any other example of this generating an interesting set of statistics?**
>
> It is possible to use functions other than polynomials, but the challenge with doing this is that the application of the Langevin Stein operator, which is required for the zero expectation property, modifies the form of the function. This means we may not obtain interpretable forms for other functions like we do for polynomial functions and Gaussian distributions. As an example, if one were to consider $g(x) = \sin(x)$, then after the application of a second-order Langevin Stein operator for a one-dimensional unit Gaussian target, the function would become $\mathcal{A}_x^{(2)} g(x) = -\sin(x) - x \cos(x)$. A discrepancy based on $g(x) = \sin(x)$ would therefore determine whether $\mathbb{E}_Q[\sin(x) + x \cos(x)] = \mathbb{E}_P[\sin(x) + x \cos(x)]$, which is not the same as the desired function $\sin(x)$.
>
> **For instance, if I wanted to test against the moments after applying some transform to my random variable, could I possibly concoct interesting tests this way?**
>
> This is an interesting question. Appendix D shows that applying an invertible linear transformation does not change the fact that, in the Gaussian case, the test is for the moments up to order $r$ of the original distribution. However, the statistical power can be affected by the choice of transformation. In particular, such a transformation may improve power when the posterior variances differ substantially in scale. It would be interesting to consider other transformations as well, and we will mention this in the discussion.
>
> Following up on this and a comment by Reviewer JxPo on the effect of a mean shift on the PSD statistic, we aim to empirically investigate the effect of a mean shift during the review period.
>
> **My main concern is that the paper could have benefited from further exploration of some of the ideas contained therein.**
>
> We have further explored and commented on several topics as part of this review. We would be happy to consider further exploring other aspects of the paper if you would like to provide further details.

---

> > ### Comment · Reviewer_7ab6 · 2025-04-04
> >
> > Thank you for your response. I would be interested in seeing these extensions. As my initial appraisal was already positive, I will maintain my score.

---

> > > ### Author Response · Authors · 2025-04-09
> > >
> > > Thank you for your comments and interest in seeing the new results. We have now investigated the concept of a mean-shift further. We believe this will be of interest to you and Reviewer JxPo.
> > >
> > > As previously mentioned, using a mean-shift does not affect which discrepancies PSD is theoretically capable of detecting but it may affect PSD's statistical power in doing so. Upon further investigation, we have found that PSD with $r=1$ is mean-shift invariant because it is based solely on the score function, which does not change with such a transformation. However, PSD with higher $r$ uses both the samples and the score function in the discrepancy so it can be sensitive to mean-shifts. One could inflate the value of PSD using arbitrary mean shifts. To investigate this further, we have performed empirical investigations into the effect of the transformation $\tilde{x} = x - \mu_Q$, where $\mu_Q$ is the mean of $Q$. We believe this is the most sensible and practical mean shift to consider. We consider two cases based on a Gaussian $P$ with unit covariance, $N=100$ and $d=5$. We are interested in the case where the mean of $P$ ($\mu_P$) is not the same across all dimensions, since we believe this is where the impact on statistical power will be greatest. For this reason, we consider two cases, (1) $\mu_P = (a,0,\ldots,0)$ and (2) $\mu_P = (0,a,\ldots,a)$. The discrepancy will be in first dimension in both cases (details below), so we are interested in how the relative scales of the means that are correctly specified versus misspecified affect the results.
> > >
> > > The table below shows the estimated statistical power based on 200 independent simulations. We consider both case (1) and case (2) in situations where the discrepancy is in the (first) mean ($\mu_Q = \mu_P + 0.5e_1$) or in the (first) variance ($\Sigma_Q[1,1] = \Sigma_P[1,1]  + 0.5$). A "t" in front of the discrepancy indicates we have performed a mean-shift reparameterisation.
> > >
> > > | Case | Misspecified | Discrepancy | $a=-1.5$ | $a=-1.0$ | $a=-0.5$ | $a=0.0$  | $a=0.5$  |
> > > |------|------------------|-----------------|------|------|------|------|------|
> > > | 1    | Mean         | PSD$_2$     | 0.70 | 0.11 | 0.06 | 0.16 | 0.87 |
> > > |      |              | tPSD$_2$    | 0.98 | 0.22 | 0.06 | 0.13 | 0.84 |
> > > |      |              | PSD$_3$     | 0.71 | 0.26 | 0.06 | 0.18 | 0.76 |
> > > |      |              | tPSD$_3$    | 0.92 | 0.30 | 0.05 | 0.19 | 0.74 |
> > > | 2    | Mean         | PSD$_2$     | 1.00 | 1.00 | 0.76 | 0.20 | 0.08 |
> > > |      |              | tPSD$_2$    | 1.00 | 1.00 | 0.84 | 0.14 | 0.06 |
> > > |      |              | PSD$_3$     | 1.00 | 0.96 | 0.72 | 0.26 | 0.06 |
> > > |      |              | tPSD$_3$    | 1.00 | 1.00 | 0.82 | 0.25 | 0.10 |
> > > | 1    | Variance     | PSD$_2$     | 1.00 | 0.96 | 0.78 | 0.55 | 0.32 |
> > > |      |              | tPSD$_2$    | 1.00 | 0.98 | 0.84 | 0.54 | 0.32 |
> > > |      |              | PSD$_3$     | 0.96 | 0.92 | 0.70 | 0.36 | 0.17 |
> > > |      |              | tPSD$_3$    | 0.97 | 0.90 | 0.70 | 0.32 | 0.17 |
> > > | 2    | Variance     | PSD$_2$     | 0.97 | 0.83 | 0.62 | 0.39 | 0.27 |
> > > |      |              | tPSD$_2$    | 1.00 | 0.98 | 0.88 | 0.56 | 0.34 |
> > > |      |              | PSD$_3$     | 0.83 | 0.65 | 0.44 | 0.29 | 0.16 |
> > > |      |              | tPSD$_3$    | 0.98 | 0.90 | 0.70 | 0.36 | 0.16 |
> > >
> > > The results demonstrate that the original mean scaling affects the statistical power. The performance with the mean-shift reparamerisation is generally similar to, if not slightly better than, the performance with no reparameterisation.
> > >
> > > Importantly, the choice of parameterisation is a problem that affects Stein discrepancies more generally. KSD with radial kernels, i.e. kernels that are functions of $\|\| x-y \|\|$ like the Gaussian kernel, are mean-shift invariant. However, they are still sensitive to other reparameterisations, such as whitening.
> > >
> > > Determining the optimal parameterisation for Stein discrepancies is an open problem and an interesting point for further research. We will highlight this and the sensitivity of PSD to mean-shifts when $r\geq 2$ in the discussion.

---

### Official Review · Reviewer_iLVd · 2025-03-21

**Overall Recommendation:** 4

**Summary:**

The authors introduce a variant of Stein discrepancy which uses bounded degree polynomials as a Stein set. Computing the proposed Polynomial Stein Discrepancy (PSD) is straightforward using evaluations of the target score $\nabla \log P(x)$ and samples from the proposal distribution $Q$. For degree $r$, evaluating PSD given $n$ samples takes time $O(n {d+r \choose d}) \sim O(n d^r)$, whereas Kernel Stein Discrepancy requires $O(n^2 d)$ steps, which can be prohibitive for large sample sizes.

To validate PSD, the authors compare it to KSD with IMQ and Gaussian kernels, as well as two competing linear time discrepancy metrics (RFSD and Gauss FSSD-opt). For goodness of fit testing, the PSD demonstrates exceptional performance relative to competing methods in distinguishing $P=\mathcal{N}(0, I_d)$ from a few different proof-of-concept $Q$ distributions, even outperforming KSD. The PSD is also competitive with KSD and outperforms other linear time methods for goodness-of-fit testing for the samples generated by an RBM. Finally, for parameter tuning in a simple test-case LMC method, the optimal parameter choice under PSD agrees with that of KSD, while being significantly cheaper to compute.

## update after rebuttal

I appreciate the authors engagement with my questions and I keep my score.

**Claims And Evidence:**

The experimental claims made in this work are supported by established benchmarks. The derivations and proofs in this work are correct to the best of my knowledge.

**Essential References Not Discussed:**

N/a

**Experimental Designs Or Analyses:**

I am unfamiliar with the literature on discrepancy metrics for testing sample quality. However, to the best of my knowledge, this work has used well-established experimental baselines to demonstrate the efficacy of the proposed approach. To the best of my knowledge the comparisons made between PSD and competitors are fair.

**Methods And Evaluation Criteria:**

The experimental claims in this work are well supported. The authors compare PSD to other algorithms using established benchmarks, such as the goodness-of-fit tests used in Gorham & Mackey 2018, the RBM sampling task used in Liu et al. 2016 and Jitkrittum et al. 2017, and the bimodal Gaussian task introduced in Welling & Teh (2011).

**Other Comments Or Suggestions:**

In Appendix A, only the first sequence of equalities is necessary. The fact that $\\|\\bar{z}\\|\_2^2 = \\sup\_{\\beta \\in \\mathbb{R}^J : \\|\\beta\\|\_2 \\leq 1} \\sum\_{k=1}^J \\beta\_k \\bar{z}\_k$ is a well known property called 'self duality of $l^2$ norm' and it does not need to be re-proven.

Some very minor typos:
- Line 104 left 'PSD discrepancy' should just be 'PSD'
- Line 95 right should read $\mathbb{E}_{X \sim P} [(\mathcal{A}_x^{(2)} g)(X)]$, inside the expectation is a random variable

**Other Strengths And Weaknesses:**

One strength of this work is that its writing is extremely clear and easy to understand. Also, the authors give many practical insights, e.g. "we recommend the bootstrap test in general because we empirically find that it has higher power than the asymptotic test using samples from $Q$" "we have also found empirically that the power of the goodness-of-fit test based on RFSD is reduced when direct sampling from $P$ is infeasible," etc.

**Questions For Authors:**

When does KSD outperform PSD? In Figure 1, there are no such examples, and in Table 1 setting $r > 1$ is sufficient to match KSD. What are some practical examples of non-moment-based perturbations that KSD can detect but not PSD?

**Relation To Broader Scientific Literature:**

This paper belongs to the literature on Stein discrepancies and goodness-of-fit testing. The proposed method is a conceptually simple instantiation of this idea which is shown to outperform other linear-time methods in the literature.

**Theoretical Claims:**

I checked the derivation of PSD, the proof of Proposition 3.2, and the proofs detailed in Appendices A and B.

---

> ### Author Rebuttal · Authors · 2025-04-01
>
> Thank you for your time reviewing our manuscript and for your valuable comments and suggestions.
>
> We will correct the typographical issues and the appendix as suggested.
>
> **When does KSD outperform PSD? In Figure 1, there are no such examples, and in Table 1 setting $r>1$ is sufficient to match KSD. What are some practical examples of non-moment-based perturbations that KSD can detect but not PSD?**
>
> Probability distributions are completely determined by their moment generating functions, provided it exists (a necessary condition is that all moments are finite). Therefore, we can potentially expect the KSD to outperform PSD for target distributions lacking well-defined moments (e.g. those with heavy tails). We comment on this further in response to Reviewer 8qrD. We can also expect KSD to outperform PSD of order $r$ when the discrepancies lie in moments higher than the $r$th order moment. For example, KSD outperforms PSD with $r=1,2,3$ for the student-t and Laplace examples in Figure 1 since the discrepancy is in the kurtosis ($4$th order moment). We will add details of when we can expect KSD to outperform PSD to the discussion.

---

> > ### Comment · Reviewer_iLVd · 2025-04-03
> >
> > Thank you for the informative answer as to when KSD outperforms PSD. This helps me understand better the method. I plan to keep my score.

---

### Decision · Program_Chairs · 2025-05-01

**Decision:**

Accept (poster)

**Comment:**

The paper proposes a novel discrepancy metric, the Polynomial Stein Discrepancy (PSD), as a scalable alternative to the Kernel Stein Discrepancy (KSD) for assessing the quality of samples from approximate Bayesian inference algorithms. By leveraging bounded-degree polynomial test functions, the method enables linear-time computation, making it appealing for large-scale applications. The paper combines theoretical grounding for Gaussian targets with empirical validation on both synthetic and real-world scenarios.
The strengths of the paper are its:
* Theoretical Contributions: The PSD’s ability to test moment convergence under Gaussian targets is formally established (Proposition 3.2), and the paper offers a nuanced discussion of its limitations and domain of applicability.
* Practical Impact: Reviewers appreciate the computational efficiency of PSD and its utility in tasks like hyperparameter tuning for sampling algorithms such as SG-MCMC.
There were, however, some shared limitations and areas for improvement noted:
* Generality: The method is strongly grounded in the Gaussian case. While it performs well empirically on some non-Gaussian targets, its theoretical robustness outside this setting remains unclear both empirically (ie on more examples) and theoretically. This is clearly the biggest weakness of the paper.
* Theoretical Framing: Reviewer 8qrD argues that the framing via the Bernstein-von Mises (BvM) limit is somewhat overstated and potentially misleading, and recommends grounding the claims more directly in the Gaussian setting. The authors agreed and clarified this in the rebuttal.
* Comparisons: Reviewer ExjZ and others note that additional comparisons to other discrepancy-based and moment-based evaluation strategies (e.g., variational inference diagnostics) could strengthen the practical recommendations.

All reviewers had a positive opinion. with three giving a score of 4 (accept) and one giving a 3 (weak accept). While its main results are for Gaussian targets, the combination of theoretical guarantees and empirical performance makes it a valuable addition to the toolbox for assessing sample quality in approximate inference. The authors' responses addressed most of the reviewers' concerns, and the discussion around limitations and extensions (e.g., some non-Gaussian targets, choice of Stein set) was thoughtful and transparent. This paper is likely to be of interest to both theoretical and applied audiences in the Monte Carlo and Bayesian communities. Final Recommendation: Accept